# Optimistic Distributionally Robust Optimization for Nonparametric Likelihood Approximation

**Viet Anh Nguyen**     **Soroosh Shafieezadeh-Abadeh**
École Polytechnique Fédérale de Lausanne, Switzerland
{viet-anh.nguyen, soroosh.shafiee}@epfl.ch

**Man-Chung Yue**
The Hong Kong Polytechnic University, Hong Kong
manchung.yue@polyu.edu.hk

**Daniel Kuhn**
École Polytechnique Fédérale de Lausanne, Switzerland
daniel.kuhn@epfl.ch

**Wolfram Wiesemann**
Imperial College Business School, United Kingdom
ww@imperial.ac.uk

## Abstract

The likelihood function is a fundamental component in Bayesian statistics. How-ever, evaluating the likelihood of an observation is computationally intractable in many applications. In this paper, we propose a non-parametric approximation of the likelihood that identifies a probability measure which lies in the neighbor-hood of the nominal measure and that maximizes the probability of observing the given sample point. We show that when the neighborhood is constructed by the Kullback-Leibler divergence, by moment conditions or by the Wasserstein distance, then our *optimistic likelihood* can be determined through the solution of a convex optimization problem, and it admits an analytical expression in particular cases. We also show that the posterior inference problem with our optimistic likelihood approximation enjoys strong theoretical performance guarantees, and it performs competitively in a probabilistic classification task.

## 1   Introduction

Bayesian statistics is a versatile mathematical framework for estimation and inference, with appli-cations in bioinformatics [1], computational biology [40, 41], neuroscience [50], natural language processing [24, 34], computer vision [21, 25], robotics [13], machine learning [28, 46], etc. A Bayesian inference model is composed of an unknown parameter $\theta$ from a known parameter space $\Theta$, an observed sample point $x$ from a sample space $\mathcal{X} \subseteq \mathbb{R}^m$, a likelihood measure (or conditional density) $p(\cdot|\theta)$ over $\mathcal{X}$ and a prior distribution $\pi(\cdot)$ over $\Theta$. The key objective of Bayesian statistics is the computation of the posterior distribution $p(\cdot|x)$ over $\Theta$ upon observing $x$.

Unfortunately, computing the posterior is a challenging task in practice. Bayes' theorem, which relates the posterior to the prior [42, Theorem 1.31], requires the evaluation of both the likelihood function $p(\cdot|\theta)$ and the evidence $p(x)$. Evaluating the likelihood $p(\cdot|\theta)$ at an observation $x \in \mathcal{X}$ is

an intractable problem in many situations. For example, the statistical model may contain hidden variables $\zeta$, and the likelihood $p(x|\theta)$ can only be computed by marginalizing out the hidden variables $p(x|\theta) = \int p(x, \zeta|\theta)\mathrm{d}\zeta$ [32, pp. 322]. In the g-and-k model, the density function does not exist in closed form and can only be expressed in terms of the derivatives of quantile functions, which implies that $p(x|\theta)$ needs to be computed numerically for each individual observation $x$ [18]. Likewise, evaluating the evidence $p(x)$ is intractable whenever the evaluation of the likelihood $p(x|\theta)$ is. To avoid calculating $p(x)$ in the process of constructing the posterior, the variational Bayes approach [8] maximizes the evidence lower bound (ELBO), which is tantamount to solving

$$\min_{\mathbb{Q} \in \mathcal{Q}} \ \mathrm{KL}(\mathbb{Q} \parallel \pi) - \mathbb{E}_{\mathbb{Q}}[\log p(x|\theta)], \tag{1}$$

where $\mathrm{KL}(\mathbb{Q} \parallel \pi)$ denotes the Kullback-Leibler (KL) divergence from $\mathbb{Q}$ to $\pi$. One can show that if the feasible set $\mathcal{Q}$ contains all probability measures supported on $\Theta$, then the optimal solution $\mathbb{Q}^\star$ of (1) coincides with the true posterior distribution. Consequently, inferring the posterior is equivalent to solving the convex optimization problem (1) that depends only on the prior distribution $\pi$ and the likelihood $p(x|\theta)$. There are scalable algorithms to solve the ELBO maximization problem [19], and the variational Bayes approach has been successfully applied in inference tasks [15, 16], reinforcement learning [20, 30], dimensionality reduction [33] and training deep neural networks [22]. Nevertheless, the variational Bayes approach requires both perfect knowledge and a tractable representation of the likelihood $p(x|\theta)$, which is often not available in practice.

While the likelihood $p(x|\theta)$ may be intractable to compute, we can approximate $p(x|\theta)$ from available data in many applications. For example, in the classification task where $\Theta = \{\theta_1, \ldots, \theta_C\}$ denotes the class labels, the class conditional probabilities $p(x|\theta_i)$ and the prior distribution $\pi(\theta_i)$ can be inferred from the training data, and a probabilistic classifier can be constructed by assigning $x$ to each class randomly under the posterior distribution [7, pp. 43]. Approximating the intractable likelihood from available samples is also the key ingredient of approximate Bayesian computation (ABC), a popular statistical method for likelihood-free inference that has gained widespread success in various fields [2, 12, 47]. The sampling-based likelihood algorithm underlying ABC assumes that we have access to a simulation device that can generate $N$ i.i.d. samples $\widehat{x}_1, \ldots, \widehat{x}_N$ from $p(\cdot|\theta)$, and it approximates the likelihood $p(x|\theta)$ by the surrogate $p_h(x|\theta)$ defined as

$$p_h(x|\theta) \ = \ \int_{\mathcal{X}} K_h\left(d(x, \widehat{x})\right) p(\widehat{x}|\theta)\mathrm{d}\widehat{x} \ \approx \ \frac{1}{N}\sum_{j=1}^{N} K_h\left(d(x, \widehat{x}_j)\right), \tag{2}$$

where $K_h$ is a kernel function with kernel width $h$, $d(\cdot, \cdot)$ is a distance on $\mathcal{X}$, and the approximation is due to the reliance upon finitely many samples [37, 39].

In this paper, we propose an alternative approach to approximate the likelihood $p(x|\theta)$. We assume that the sample space $\mathcal{X}$ is countable, and hence $p(\cdot|\theta)$ is a probability mass function. We model the decision maker's nominal belief about $p(\cdot|\theta)$ by a nominal probability mass function $\widehat{\nu}_\theta$ supported on $\mathcal{X}$, which in practice typically represents the empirical distribution supported on the (possibly simulated) training samples. We then approximate the likelihood $p(x|\theta)$ by the optimal value of the following non-parametric *optimistic likelihood* problem

$$\sup_{\nu \in \mathbb{B}_\theta(\widehat{\nu}_\theta)} \ \nu(x), \tag{3}$$

where $\mathbb{B}_\theta(\widehat{\nu}_\theta)$ is a set that contains all probability mass functions in the vicinity of $\widehat{\nu}_\theta$. In the distributionally robust optimization literature, the set $\mathbb{B}_\theta(\widehat{\nu}_\theta)$ is referred to as the ambiguity set [3, 29, 49]. In contrast to the distributionally robust optimization paradigm, which would look for a worst-case measure that *minimizes* the probability of observing $x$ among all measures contained in $\mathbb{B}_\theta(\widehat{\nu}_\theta)$, the optimistic likelihood problem (3) determines a best-case measure that *maximizes* this quantity. Thus, problem (3) is closely related to the literature on practicing optimism upon facing ambiguity, which has been shown to be beneficial in multi-armed bandit problems [10], planning [31], classification [6], image denoising [17], Bayesian optimization [9, 45], etc.

The choice of the set $\mathbb{B}_\theta(\widehat{\nu}_\theta)$ in (3) directly impacts the performance of the optimistic likelihood approach. In the limiting case where $\mathbb{B}_\theta(\widehat{\nu}_\theta)$ approaches a singleton $\{\widehat{\nu}_\theta\}$, the optimistic likelihood problem recovers the nominal estimate $\widehat{\nu}_\theta(x)$. Since this approximation is only reasonable when $\widehat{\nu}_\theta(x) > 0$, which is often violated when $\widehat{\nu}_\theta$ is estimated from few training samples, a strictly positive size of $\mathbb{B}_\theta(\widehat{\nu}_\theta)$ is preferred. Ideally, the shape of $\mathbb{B}_\theta(\widehat{\nu}_\theta)$ is chosen so that problem (3)

is computationally tractable and at the same time offers a promising approximation quality. We explore in this paper three different constructions of $\mathbb{B}_\theta(\widehat{\nu}_\theta)$: the Kullback-Leibler divergence [3], a description based on moment conditions [14, 27] and the Wasserstein distance [23, 35, 38, 43, 44].

The contributions of this paper may be summarized as follows.

1. We show that when $\mathbb{B}_\theta(\widehat{\nu}_\theta)$ is constructed using the KL divergence, the optimistic likelihood (3) reduces to a finite convex program, which in specific cases admits an analytical solution. However, this approach does not satisfactorily approximate $p(x|\theta)$ for previously unseen samples $x$.

2. We demonstrate that when $\mathbb{B}_\theta(\widehat{\nu}_\theta)$ is constructed using moment conditions, the optimistic likelihood (3) can be computed in closed form. However, since strikingly different distributions can share the same lower-order moments, this approach is often not flexible enough to accurately capture the tail behavior of $\widehat{\nu}_\theta$.

3. We show that when $\mathbb{B}_\theta(\widehat{\nu}_\theta)$ is constructed using the Wasserstein distance, the optimistic likelihood (3) coincides with the optimal value of a linear program that can be solved using a greedy heuristics. Interestingly, this variant of the optimistic likelihood results in a likelihood approximation whose decay pattern resembles that of an exponential kernel approximation.

4. We use our optimistic likelihood approximation in the ELBO problem (1) for posterior inference. We prove that the resulting posterior inference problems under the KL divergence and the Wasserstein distance enjoy strong theoretical guarantees, and we illustrate their promising empirical performance in numerical experiments.

While this paper focuses on the non-parametric approximation of the likelihood $p(x|\theta)$, we emphasize that the optimistic likelihood approach can also be applied in the parametric setting. More specifically, if $p(\cdot|\theta)$ belongs to the family of Gaussian distributions, then the optimistic likelihood approximation can be solved efficiently using geodesically convex optimization [36].

The remainder of the paper is structured as follows. We study the optimistic likelihood problem under the KL ambiguity set, under moment conditions and under the Wasserstein distance in Sections 2–4, respectively. Section 5 provides a performance guarantee for the posterior inference problem using our optimistic likelihood. All proofs and additional material are relegated to the Appendix. In Sections 2–4, the development of the theoretical results is generic, and hence the dependence of $\widehat{\nu}_\theta$ and $\mathbb{B}_\theta(\widehat{\nu}_\theta)$ on $\theta$ is omitted to avoid clutter.

**Notation.** We denote by $\mathcal{M}(\mathcal{X})$ the set of all probability mass functions supported on $\mathcal{X}$, and we refer to the support of $\nu \in \mathcal{M}(\mathcal{X})$ as $\operatorname{supp}(\nu)$. For any $z \in \mathcal{X}$, $\delta_z$ is the delta-Dirac measure at $z$. For any $N \in \mathbb{N}_+$, we use $[N]$ to denote the set $\{1, \ldots, N\}$. $\mathbb{1}_x(\cdot)$ is the indicator function at $x$, i.e., $\mathbb{1}_x(\xi) = 1$ if $\xi = x$, and $\mathbb{1}_x(\xi) = 0$ otherwise.

## 2 Optimistic Likelihood using the Kullback-Leibler Divergence

We first consider the optimistic likelihood problem where the ambiguity set is constructed using the KL divergence. The KL divergence is the starting point of the ELBO maximization problem (1), and thus it is natural to explore its potential in our likelihood approximation.

**Definition 2.1** (KL divergence). Let $\nu_1, \nu_2$ be two probability mass functions on $\mathcal{X}$ such that $\nu_1$ is absolutely continuous with respect to $\nu_2$. The KL divergence between $\nu_1$ and $\nu_2$ is defined as

$$\mathrm{KL}(\nu_1 \parallel \nu_2) \triangleq \sum_{z \in \mathcal{X}} f\left(\nu_1(z)/\nu_2(z)\right) \nu_2(z),$$

where $f(t) = t \log(t) - t + 1$.

We now consider the KL divergence ball $\mathbb{B}_{\mathrm{KL}}(\widehat{\nu}, \varepsilon)$ centered at the empirical distribution $\widehat{\nu}$ with radius $\varepsilon \geq 0$, that is,

$$\mathbb{B}_{\mathrm{KL}}(\widehat{\nu}, \varepsilon) = \{\nu \in \mathcal{M}(\mathcal{X}) : \mathrm{KL}(\widehat{\nu} \parallel \nu) \leq \varepsilon\}. \tag{4}$$

Moreover, we assume that the nominal distribution $\widehat{\nu}$ is supported on $N$ distinct points $\widehat{x}_1, \ldots, \widehat{x}_N$, that is, $\widehat{\nu} = \sum_{j \in [N]} \widehat{\nu}_j \delta_{\widehat{x}_j}$ with $\widehat{\nu}_j > 0 \ \forall j \in [N]$ and $\sum_{j \in [N]} \widehat{\nu}_j = 1$.

The set $\mathbb{B}_{\mathrm{KL}}(\widehat{\nu}, \varepsilon)$ is not weakly compact because $\mathcal{X}$ can be unbounded, and thus the existence of a probability measure that optimizes the optimistic likelihood problem (3) over the feasible set

$\mathbb{B}_{\mathrm{KL}}(\widehat{\nu}, \varepsilon)$ is not immediate. The next proposition asserts that the optimal solution exists, and it provides structural insights about the support of the optimal measure.

**Proposition 2.2** (Existence of optimizers; KL ambiguity). *For any $\varepsilon \geq 0$ and $x \in \mathcal{X}$, there exists a measure $\nu_{\mathrm{KL}}^{\star} \in \mathbb{B}_{\mathrm{KL}}(\widehat{\nu}, \varepsilon)$ such that*

$$\sup_{\nu \in \mathbb{B}_{\mathrm{KL}}(\widehat{\nu}, \varepsilon)} \nu(x) = \nu_{\mathrm{KL}}^{\star}(x) \tag{5}$$

*Moreover, $\nu_{\mathrm{KL}}^{\star}$ is supported on at most $N + 1$ points satisfying $\mathrm{supp}(\nu_{\mathrm{KL}}^{\star}) \subseteq \mathrm{supp}(\widehat{\nu}) \cup \{x\}$.*

Proposition 2.2 suggests that the optimistic likelihood problem (5), inherently an *in*finite dimensional problem whenever $\mathcal{X}$ is infinite, can be formulated as a finite dimensional problem. The next theorem provides a finite convex programming reformulation of (5).

**Theorem 2.3** (Optimistic likelihood; KL ambiguity). *For any $\varepsilon \geq 0$ and $x \in \mathcal{X}$,*

- *if $x \in \mathrm{supp}(\widehat{\nu})$, then problem (5) can be reformulated as the finite convex optimization problem*

$$\sup_{\nu \in \mathbb{B}_{\mathrm{KL}}(\widehat{\nu}, \varepsilon)} \nu(x) = \max \left\{ \textstyle\sum_{j \in [N]} y_j \mathbb{1}_x(\widehat{x}_j) : y \in \mathbb{R}_{++}^N, \; \textstyle\sum_{j \in [N]} \widehat{\nu}_j \log(\widehat{\nu}_j / y_j) \leq \varepsilon, \; e^{\top} y = 1 \right\},$$

  *where $e$ is the vector of all ones;*

- *if $x \notin \mathrm{supp}(\widehat{\nu})$, then problem (5) has the optimal value $1 - \exp(-\varepsilon)$.*

Theorem 2.3 indicates that the determining factor in the KL optimistic likelihood approximation is whether the observation $x$ belongs to the support of the nominal measure $\widehat{\nu}$ or not. If $x \notin \mathrm{supp}(\widehat{\nu})$, then the optimal value of (5) does not depend on $x$, and the KL divergence approach assigns a flat likelihood. Interestingly, in Appendix B.2 we prove a similar result for the wider class of $f$-divergences, which contains the KL divergence as a special case. While this flat likelihood behavior may be useful in specific cases, one would expect the relative distance of $x$ to the atoms of $\widehat{\nu}$ to influence the optimal value of the optimistic likelihood problem, similar to the neighborhood-based intuition reflected in the kernel approximation approach. Unfortunately, the lack of an underlying metric in its definition implies that the $f$-divergence family cannot capture this intuition, and thus $f$-divergence ambiguity sets are not an attractive option to approximate the likelihood of an observation $x$ that does not belong to the support of the nominal measure $\widehat{\nu}$.

**Remark 2.4** (On the order of the measures). An alternative construction of the KL ambiguity set, which has been widely used in the literature [3], is

$$\widehat{\mathbb{B}}_{\mathrm{KL}}(\widehat{\nu}, \varepsilon) = \{\nu \in \mathcal{M}(\mathcal{X}) : \mathrm{KL}(\nu \parallel \widehat{\nu}) \leq \varepsilon\},$$

where the two measures $\nu$ and $\widehat{\nu}$ change roles. However, in this case the KL divergence imposes that all $\nu \in \widehat{\mathbb{B}}_{\mathrm{KL}}(\widehat{\nu}, \varepsilon)$ are absolutely continuous with respect to $\widehat{\nu}$. In particular, if $x \notin \mathrm{supp}(\widehat{\nu})$, then $\nu(x) = 0$ for all $\nu \in \widehat{\mathbb{B}}_{\mathrm{KL}}(\widehat{\nu}, \varepsilon)$, and $\widehat{\mathbb{B}}_{\mathrm{KL}}(\widehat{\nu}, \varepsilon)$ is not able to approximate the likelihood of $x$ in a meaningful way.

## 3 Optimistic Likelihood using Moment Conditions

In this section we study the optimistic likelihood problem (3) when the ambiguity set $\mathbb{B}(\widehat{\nu})$ is specified by moment conditions. For tractability purposes, we focus on ambiguity sets $\mathbb{B}_{\mathrm{MV}}(\widehat{\nu})$ that contain all distributions which share the same mean $\widehat{\mu}$ and covariance matrix $\widehat{\Sigma} \in \mathbb{S}_{++}^m$ with the nominal distribution $\widehat{\nu}$. Formally, this moment ambiguity set $\mathbb{B}_{\mathrm{MV}}(\widehat{\nu})$ can be expressed as

$$\mathbb{B}_{\mathrm{MV}}(\widehat{\nu}) = \left\{ \nu \in \mathcal{M}(\mathcal{X}) : \; \mathbb{E}_{\nu}[\tilde{x}] = \widehat{\mu}, \; \mathbb{E}_{\nu}[\tilde{x}\tilde{x}^{\top}] = \widehat{\Sigma} + \widehat{\mu}\widehat{\mu}^{\top} \right\}.$$

The optimistic likelihood (3) over the ambiguity set $\mathbb{B}_{\mathrm{MV}}(\widehat{\nu})$ is a moment problem that is amenable to a well-known reformulation as a polynomial time solvable semidefinite program [5]. Surprisingly, in our case the optimal value of the optimistic likelihood problem is available in closed form. This result was first discovered in [26], and a proof using optimization techniques can be found in [4].

**Theorem 3.1** (Optimistic likelihood; mean-variance ambiguity [4, 26])**.** Suppose that $\widehat{\nu}$ has the mean vector $\widehat{\mu} \in \mathbb{R}^m$ and the covariance matrix $\widehat{\Sigma} \in \mathbb{S}_{++}^m$. For any $x \in \mathcal{X}$, the optimistic likelihood problem (3) over the moment ambiguity set $\mathbb{B}_{\mathrm{MV}}(\widehat{\nu})$ has the optimal value

$$\sup_{\nu \in \mathbb{B}_{\mathrm{MV}}(\widehat{\nu})} \nu(x) = \frac{1}{1 + (x - \widehat{\mu})^\top \widehat{\Sigma}^{-1}(x - \widehat{\mu})} \in (0, 1]. \tag{6}$$

The optimal value (6) of the optimistic likelihood problem depends on the location of the observed sample point $x$, and hence the moment ambiguity set captures the behavior of the likelihood function in a more realistic way than the KL divergence ambiguity set from Section 2. Moreover, the moment ambiguity set $\mathbb{B}_{\mathrm{MV}}(\widehat{\nu})$ does not depend on any hyper-parameters that need to be tuned. However, since the construction of $\mathbb{B}_{\mathrm{MV}}(\widehat{\nu})$ only relies on the first two moments of the nominal distribution $\widehat{\nu}$, it fails to accurately capture the tail behavior of $\widehat{\nu}$, see Appendix B.3. This motivates us to look further for an ambiguity set that faithfully accounts for the tail behavior of $\widehat{\nu}$.

## 4 Optimistic Likelihood using the Wasserstein Distance

We now study a third construction for the ambiguity set $\mathbb{B}(\widehat{\nu})$, which is based on the type-1 Wasserstein distance (also commonly known as the Monge-Kantorovich distance), see [48]. Contrary to the KL divergence, the Wasserstein distance inherently depends on the ground metric of the sample space $\mathcal{X}$.

**Definition 4.1** (Wasserstein distance)**.** The type-1 Wasserstein distance between two measures $\nu_1, \nu_2 \in \mathcal{M}(\mathcal{X})$ is defined as

$$\mathbb{W}(\nu_1, \nu_2) \triangleq \inf_{\lambda \in \Lambda(\nu_1, \nu_2)} \mathbb{E}_\lambda \left[ d(x_1, x_2) \right],$$

where $\Lambda(\nu_1, \nu_2)$ denotes the set of all distributions on $\mathcal{X} \times \mathcal{X}$ with the first and second marginal distributions being $\nu_1$ and $\nu_2$, respectively, and $d$ is the ground metric of $\mathcal{X}$.

The Wasserstein ball $\mathbb{B}_{\mathrm{W}}(\widehat{\nu}, \varepsilon)$ centered at the nominal distribution $\widehat{\nu}$ with radius $\varepsilon \geq 0$ is

$$\mathbb{B}_{\mathrm{W}}(\widehat{\nu}, \varepsilon) = \{ \nu \in \mathcal{M}(\mathcal{X}) : \mathbb{W}(\nu, \widehat{\nu}) \leq \varepsilon \}. \tag{7}$$

We first establish a structural result for the optimistic likelihood problem over the Wasserstein ambiguity set. This is the counterpart to Proposition 2.2 for the KL divergence.

**Proposition 4.2** (Existence of optimizers; Wasserstein ambiguity)**.** For any $\varepsilon \geq 0$ and $x \in \mathcal{X}$, there exists a measure $\nu_{\mathrm{W}}^\star \in \mathbb{B}_{\mathrm{W}}(\widehat{\nu}, \varepsilon)$ such that

$$\sup_{\nu \in \mathbb{B}_{\mathrm{W}}(\widehat{\nu}, \varepsilon)} \nu(x) = \nu_{\mathrm{W}}^\star(x). \tag{8}$$

Furthermore, $\nu_{\mathrm{W}}^\star$ is supported on at most $N + 1$ points satisfying $\mathrm{supp}(\nu_{\mathrm{W}}^\star) \subseteq \mathrm{supp}(\widehat{\nu}) \cup \{x\}$.

Leveraging Proposition 4.2, we can show that the optimistic likelihood estimate over the Wasserstein ambiguity set coincides with the optimal value of a linear program whose number of decision variables equals the number of atoms $N$ of the nominal measure $\widehat{\nu}$.

**Theorem 4.3** (Optimistic likelihood; Wasserstein ambiguity)**.** For any $\varepsilon \geq 0$ and $x \in \mathcal{X}$, problem (8) is equivalent to the linear program

$$\sup_{\nu \in \mathbb{B}_{\mathrm{W}}(\widehat{\nu}, \varepsilon)} \nu(x) = \max \left\{ \sum_{j \in [N]} T_j : T \in \mathbb{R}_+^N, \ \sum_{j \in [N]} d(x, \widehat{x}_j) \, T_j \leq \varepsilon, \ T_j \leq \widehat{\nu}_j \ \forall j \in [N] \right\}. \tag{9}$$

The currently best complexity bound for solving a general linear program with $N$ decision variables is $\mathcal{O}(N^{2.37})$ [11], which may be prohibitive when $N$ is large. Fortunately, the linear program (9) can be solved to optimality using a greedy heuristics in quasilinear time.

**Proposition 4.4** (Optimal solution via greedy heuristics)**.** The linear program (9) can be solved to optimality by a greedy heuristics in time $\mathcal{O}(N \log N)$.

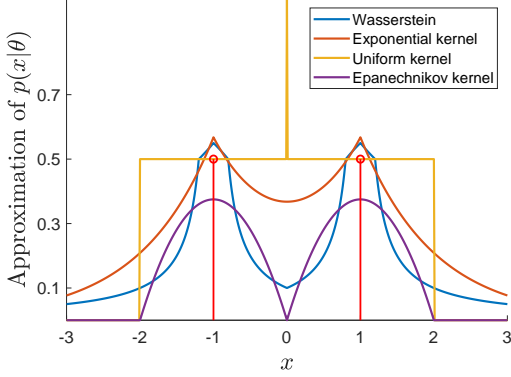

Figure 1: Comparison between the Wasserstein approximation ($\varepsilon = 0.2$) and the sample average kernel approximations ($h = 1$) of $p(x|\theta)$.

**Example 4.5** (Qualitative comparison with kernel methods). Let $m = 1$, $d(x, \widehat{x}) = \|x - \widehat{x}\|_1$ and $\widehat{\nu} = 0.5\delta_{-1} + 0.5\delta_1$. Figure 1 compares the approximation of $p(x|\theta)$ by the Wasserstein optimistic likelihood with those of the finite sample kernel approximations (2) with $K_h(u) = K(h^{-1}u)$, where the Kernel $K$ is exponential with $K(y) = \exp(-y)$, uniform with $K(y) = \mathbb{1}[|y| \leq 1]$ or Epanechnikov with $K(y) = 3/4(1 - y^2)\mathbb{1}[|y| \leq 1]$. While both the uniform and the Epachnechnikov kernel may produce an approximation value of 0 when $x$ is far away from the support of $\widehat{\nu}$, the Wasserstein approximation always returns a positive likelihood when $\varepsilon > 0$ (see Corollary A.2). Qualitatively, the Wasserstein approximation exhibits a decay pattern similar to that of the finite sample average exponential kernel approximation.

On one hand, the similarity between the optimistic likelihood over the Wasserstein ambiguity set and the exponential kernel approximation suggests that the kernel approximation can potentially be interpreted in the light of our optimistic distributionally robust optimization framework. On the other hand, and perhaps more importantly, this similarity suggests that there are possibilities to design novel and computationally efficient kernel-like approximations using advanced optimization techniques. Even though the assumption that $p(\cdot|\theta)$ is a probability mass function is fundamental for our approximation, we believe that our approach can be utilized in the ABC setting even when $p(\cdot|\theta)$ is a probability density function. We leave these ideas for future research.

Appendix B.3 illustrates further how the Wasserstein ambiguity set offers a better tail approximation of the nominal measure $\widehat{\nu}$ than the ambiguity set based on moment conditions. Interestingly, the Wasserstein approximation can also be generalized to approximate the log-likelihood of a batch of i.i.d. observations, see Appendix B.4

## 5  Application to the ELBO Problem

Motivated by the fact that the likelihood $p(x|\theta)$ is intractable to compute in many practical applications, we use our optimistic likelihood approximation (3) as a surrogate for $p(x|\theta)$ in the ELBO problem (1). In this section, we will focus on the KL divergence and the Wasserstein ambiguity sets, and we will impose the following assumptions.

**Assumption 5.1** (Finite parameter space). We assume that $\Theta = \{\theta_1, \ldots, \theta_C\}$ for some $C \geq 2$.

**Assumption 5.2** (I.i.d. sampling and empirical distribution). For every $i \in [C]$, we have $N_i$ i.i.d. samples $\widehat{x}_{ij}$, $j \in [N_i]$, from the conditional probability $p(\cdot|\theta_i)$. Furthermore, each nominal distribution $\widehat{\nu}_i$ is given by the empirical distribution $\widehat{\nu}_i^{N_i} = N_i^{-1} \sum_{j \in [N_i]} \delta_{\widehat{x}_{ij}}$ on the samples $\widehat{x}_{ij}$.

Assumption 5.1 is necessary for our approach because we approximate $p(x|\theta)$ separately for every $\theta \in \Theta$. Under this assumption, the prior distribution $\pi$ can be expressed by the $C$-dimensional vector $\pi \in \mathbb{R}_+$, and the ELBO program (1) becomes the finite-dimensional convex optimization problem

$$\mathcal{J}^{\text{true}} = \min_{q \in \mathcal{Q}} \sum_{i \in [C]} q_i(\log q_i - \log \pi_i) - \sum_{i \in [C]} q_i \log p(x|\theta_i), \tag{10}$$

where by a slight abuse of notation, $\mathcal{Q}$ is now a subset of the $C$-dimensional simplex. Assumption 5.2, on the other hand, is a standard assumption in the nonparametric setting, and it allows us to study the statistical properties of our optimistic likelihood approximation.

We approximate $p(x|\theta_i)$ for each $\theta_i$ by the optimal value of the optimistic likelihood problem (3):

$$p(x|\theta_i) \approx \sup_{\nu_i \in \mathbb{B}_i^{N_i}(\widehat{\nu}_i^{N_i})} \nu_i(x) \tag{11}$$

Here, $\mathbb{B}_i^{N_i}(\widehat{\nu}_i^{N_i})$ is the KL divergence or Wasserstein ambiguity set centered at the empirical distribution $\widehat{\nu}_i^{N_i}$. Under Assumptions 5.1 and 5.2, a surrogate model of the ELBO problem (1) is then

obtained using the approximation (11) as

$$\widehat{\mathcal{J}}_{\mathbb{B}^N} = \min_{q \in \mathcal{Q}} \sum_{i \in [C]} q_i(\log q_i - \log \pi_i) - \sum_{i \in [C]} q_i \log \left( \sup_{\nu_i \in \mathbb{B}_i^{N_i}(\widehat{\nu}_i^{N_i})} \nu_i(x) \right), \qquad (12)$$

where we use $\mathbb{B}^N$ to denote the collection of ambiguity sets $\left\{ \mathbb{B}_i^{N_i}(\widehat{\nu}_i^{N_i}) \right\}_{i=1}^C$ with $N = \sum_i N_i$.

We now study the statistical properties of problem (12). We first present an asymptotic guarantee for the KL divergence. Towards this end, we define the *disappointment* as $\mathbb{P}^\infty(\mathcal{J}^{\text{true}} < \widehat{\mathcal{J}}_{\mathbb{B}^N})$.

**Theorem 5.3** (Asymptotic guarantee; KL ambiguity). Suppose that Assumptions 5.1 and 5.2 hold. For each $i \in [C]$, let $\mathbb{B}_i^{N_i}(\widehat{\nu}_i^{N_i}) = \mathbb{B}_{\text{KL}}(\widehat{\nu}_i^{N_i}, \varepsilon_i)$ for some $\varepsilon_i > 0$, and set $n \triangleq \min\{N_1, \ldots, N_C\}$. We then have

$$\limsup_{n \to \infty} \frac{1}{n} \log \mathbb{P}^\infty(\mathcal{J}^{\text{true}} < \widehat{\mathcal{J}}_{\mathbb{B}^N}) \leq -\min_{i \in [C]} \varepsilon_i < 0.$$

Theorem 5.3 shows that as the number of training samples $N_i$ for each $i \in [C]$ grows, the disappointment decays exponentially at a rate of at least $\min_i \varepsilon_i$.

We next study the statistical properties of problem (12) when each $\mathbb{B}_i^{N_i}(\widehat{\nu}_i^{N_i})$ is a Wasserstein ball. To this end, we additionally impose the following assumption, which essentially requires that the tail of each distribution $p(\cdot|\theta_i)$, $i \in [C]$, decays at an exponential rate.

**Assumption 5.4** (Light-tailed conditional distribution). For each $i \in [C]$, there exists an exponent $a_i > 1$ such that $A_i \triangleq \mathbb{E}[\exp(\|x\|^{a_i})] < \infty$, where the expectation is taken with respect to $p(\cdot|\theta_i)$.

**Theorem 5.5** (Finite sample guarantee; Wasserstein ambiguity). Suppose that Assumptions 5.1, 5.2 and 5.4 hold, and fix any $\beta \in (0,1)$. Assume that $m \neq 2$ and that $\mathbb{B}_i^{N_i}(\widehat{\nu}_i^{N_i}) = \mathbb{B}_{\mathbb{W}}(\widehat{\nu}_i^{N_i}, \varepsilon_i(\beta, C, N_i))$ for every $i \in [C]$ with

$$\varepsilon_i(\beta, C, N_i) \triangleq \begin{cases} \left( \frac{\log(k_{i1} C \beta^{-1})}{k_{i2} N_i} \right)^{1/\max\{m,2\}} & \text{if } N_i \geq \frac{\log(k_{i1}) C \beta^{-1}}{k_{i2}}, \\ \left( \frac{\log(k_{i1} C \beta^{-1})}{k_{i2} N_i} \right)^{1/a_i} & \text{if } N_i < \frac{\log(k_{i1}) C \beta^{-1}}{k_{i2}}, \end{cases}$$

and $k_{i1}, k_{i2}$ are positive constants that depend on $a_i$, $A_i$ and $m$. We then have $\mathbb{P}^N\left( \mathcal{J}^{\text{true}} < \widehat{\mathcal{J}}_{\mathbb{B}^N} \right) \leq \beta$.

Theorem 5.5 provides a finite sample guarantee for the disappointment of problem (12) under a specific choice of radii for the Wasserstein balls.

**Theorem 5.6** (Asymptotic guarantee for Wasserstein). Suppose that Assumptions 5.1, 5.2 and 5.4 hold. For each $i \in [C]$, let $\beta_{N_i} \in (0,1)$ be a sequence such that $\sum_{N_i=1}^\infty \beta_{N_i} < \infty$ and $\mathbb{B}_i^{N_i}(\widehat{\nu}_i^{N_i}) = \mathbb{B}_{\mathbb{W}}(\widehat{\nu}_i^{N_i}, \varepsilon_i(\beta_N, C, N_i))$, where $\varepsilon_i$ is defined as in Theorem 5.5. Then $\widehat{\mathcal{J}}_{\mathbb{B}^N} \to \mathcal{J}^{\text{true}}$ as $N_1, \ldots, N_C \to \infty$ almost surely.

Theorem 5.6 offers an asymptotic guarantee which asserts that as the numbers of training samples $N_i$ grow, the optimal value of (12) converges to that of the ELBO problem (10).

# 6 Numerical Experiments

We first showcase the performance guarantees from the previous section on a synthetic dataset in Section 6.1. Afterwards, Section 6.2 benchmarks the performance of the different likelihood approximations in a probabilistic classification task on standard UCI datasets. The source code, including our algorithm and all tests implemented in Python, are available from `https://github.com/sorooshafiee/Nonparam_Likelihood`.

## 6.1 Synthetic Dataset: Beta-Binomial Inference

We consider the beta-binomial problem in which the prior $\pi$, the likelihood $p(x|\theta)$, and the posterior distribution $q(\theta|x)$ have the following forms:

$$\pi(\theta) = \text{Beta}(\theta|\alpha, \beta), \quad p(x|\theta) = \text{Bin}(x|M, \theta), \quad q(\theta|x) = \text{Beta}(\theta|x + \alpha, M - x + \beta)$$

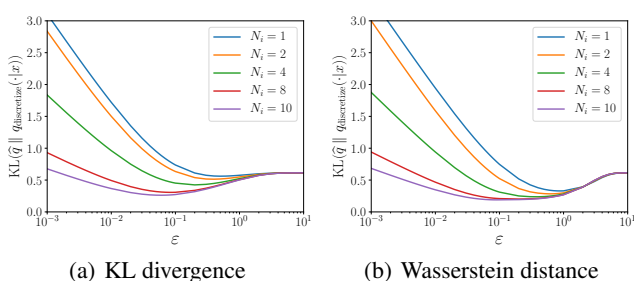
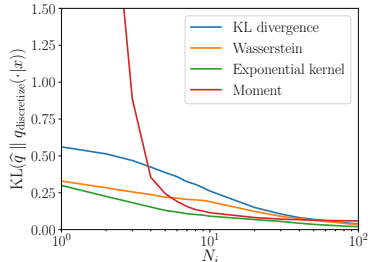

(a) KL divergence      (b) Wasserstein distance

Figure 2: Average KL divergence between $\widehat{q}$ that solves (12) and the discretized posterior $q_{\text{discretize}}(\cdot|x)$ as a function of $\varepsilon$ and $N_i$.

Figure 3: Optimally tuned performance of different approximation schemes with varying $N_i$.

We emphasize that in this setting, the posterior distribution is known in closed form, and the main goal is to study the properties of the optimistic ELBO problem (12) and the convergence of the solution of problem (12) to the true posterior distribution. We impose a uniform prior distribution $\pi$ by setting $\alpha = \beta = 1$. The finite parameter space $\Theta = \{\theta_1, \ldots, \theta_C\}$ contains $C = 20$ equidistant discrete points in the range $(0, 1)$. For simplicity, we set $N_1 = \ldots = N_C$ in this experiment.

We conduct the following experiment for different training set sizes $N_i \in \{1, 2, 4, 8, 10\}$ and different ambiguity set radii $\varepsilon$. For each parameter setting, our experiment consists of 100 repetitions. In each repetition, we randomly generate an observation $x$ from a binomial distribution with $M = 20$ trials and success probability $\theta_{\text{true}} = 0.6$. We then find the distribution $\widehat{q}$ that solves problem (12) using both the KL and the Wasserstein approximation. In a similar way, we find $\widehat{q}$ by solving (10), where $p(x|\theta)$ is approximated using the exponential kernel of the likelihood (2) with varying kernel width.

We evaluate the quality of the computed posteriors $\widehat{q}$ from the different approximations based on the KL divergences of $\widehat{q}$ to the true discretized posterior $q_{\text{discretize}}(\theta_i|x) \propto \text{Beta}(\theta_i|x + \alpha, M - x + \beta)$. Figures 2(a) and 2(b) depict the average quality of $\widehat{q}$ with different radii. One can readily see that the optimal size of the ambiguity set that minimizes $\text{KL}(\widehat{q} \,\|\, q_{\text{discretize}}(\cdot|x))$ decreases as $N_i$ increases for both the KL and the Wasserstein approximation. Figure 3 depicts the performance of the optimally tuned approximations with different sample sizes $N_i$. We notice that the optimistic likelihood over the Wasserstein ambiguity set is comparable to the exponential kernel approximation.

## 6.2 Real World Dataset: Classification

We now consider a probabilistic classification setting with $C = 2$ classes. For each class $i = 1, 2$, we have access to $N_i$ observations denoted by $\{\widehat{x}_{ij}\}_{j \in [N_i]}$. The nominal class-conditional probability distributions are the empirical measures, that is, $\widehat{\nu}_i = N_i^{-1} \sum_{j \in [N_i]} \delta_{\widehat{x}_{ij}}$ for $i = 1, 2$. The prior distribution $\pi$ is also estimated from the training data as $\pi(\theta_i) = N_i/N$, where $N = N_1 + N_2$ is the total number of training samples. Upon observing a test sample $x$, the goal is to compute the posterior distribution $\widehat{q}$ by solving the optimization problem (12) using different approximation schemes. We subsequently use the posterior $\widehat{q}$ as a probabilistic classifier. In this experiment, we exclude the KL divergence approximation because $x \notin \text{supp}(\widehat{\nu}_i)$ most of the time.

In our experiments involving the Wasserstein ambiguity set, we randomly select 75% of the available data as training set and the remaining 25% as test set. We then use the training samples to tune the radii $\varepsilon_i \in \{a\sqrt{m}10^b : a \in \{1, \ldots, 9\}, b \in \{-3, -2, -1\}\}$, $i = 1, 2$, of the Wasserstein balls by a stratified 5-fold cross validation. For the moment based approximation, there is no hyper-parameter to tune, and all data is used as training set. We compare the performance of the classifiers from our optimistic likelihood approximation against the classifier selected by the exponential kernel approximation as a benchmark.

Table 1 presents the results on standard UCI benchmark datasets. All results are averages across 10 independent trials. The table shows that our optimistic likelihood approaches often outperform the exponential kernel approximation in classification tasks.

**Acknowledgments** We gratefully acknowledge financial support from the Swiss National Science Foundation under grant BSCGI0_157733 as well as the EPSRC grants EP/M028240/1, EP/M027856/1 and EP/N020030/1.

Table 1: Average area under the precision-recall curve for various UCI benchmark datasets. Bold numbers correspond to the best performances.

|  | Exponential | Moment | Wasserstein |
|---|---|---|---|
| Banknote Authentication | 99.05 | 99.99 | **100.00** |
| Blood Transfusion | 64.91 | **71.28** | 68.23 |
| Breast Cancer | 97.58 | **99.26** | 97.99 |
| Climate Model | **93.80** | 81.94 | 93.40 |
| Cylinder | 76.74 | 75.00 | **86.23** |
| Fourclass | 99.95 | 82.77 | **100.00** |
| German Credit | 67.58 | **75.50** | 75.11 |
| Haberman | 70.82 | 70.20 | **71.10** |
| Heart | 78.77 | **86.87** | 75.86 |
| Housing | 75.62 | 81.89 | **82.04** |
| ILPD | 71.54 | **72.95** | 69.88 |
| Ionosphere | 91.02 | 97.05 | **98.79** |
| Mammographic Mass | 83.46 | 86.53 | **87.86** |
| Pima | 79.61 | **82.37** | 80.48 |
| QSAR | 84.44 | **90.85** | 90.21 |
| Seismic Bumps | 74.81 | **75.68** | 65.89 |
| Sonar | 85.66 | 83.49 | **93.85** |
| Thoracic Surgery | 54.84 | **64.73** | 56.32 |

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
