[Supplementary Material · ELBO_proof.pdf]

# Appendix A   Proofs

## A.1   Proofs of Section 2

The proof of Proposition 2.2 relies on the following auxiliary lemma, which we state first.

**Lemma A.1** (Upper semicontinuity). *For any $x \in \mathcal{X} \subset \mathbb{R}^m$, the functional $F(\nu) = \nu(x)$ is upper semicontinuous over $\mathcal{M}(\mathcal{X})$.*

*Proof.* We denote by $\mathbb{1}_x(\cdot)$ the indicator function at $x$, that is, $\mathbb{1}_x(\xi) = 1$ if $\xi = x$ and $\mathbb{1}_x(\xi) = 0$ otherwise. By definition, $F(\nu) = \int \mathbb{1}_x \mathrm{d}\nu$. Moreover, let $\{\nu_k\}_{k \in \mathbb{N}}$ be a sequence of probability measures converging weakly to $\nu \in \mathcal{M}(\mathcal{X})$. Since $\mathbb{1}_x(\cdot)$ is upper semicontinuous, the weak convergence of $\nu_k$ implies that

$$\limsup_{k \to \infty} F(\nu_k) = \limsup_{k \to \infty} \int \mathbb{1}_x \mathrm{d}\nu_k \leq \int \mathbb{1}_x \mathrm{d}\nu = F(\nu),$$

which in turn shows that the functional $F$ is upper semicontinuous. □

*Proof of Proposition 2.2.* If $\varepsilon = 0$, the ball $\mathbb{B}_{\mathrm{KL}}(\widehat{\nu}, \varepsilon)$ contains a singleton $\widehat{\nu}$ and the claim holds trivially. We can thus assume that $\varepsilon > 0$. Since $\mathbb{B}_{\mathrm{KL}}(\widehat{\nu}, \varepsilon)$ is not necessarily weakly compact, the existence of the optimal measure $\nu^\star$ is not trivial. To show that $\nu^\star$ exists, we first establish that

$$\sup_{\nu \in \mathbb{B}_{\mathrm{KL}}(\widehat{\nu}, \varepsilon)} \nu(x) = \sup_{\substack{\nu \in \mathbb{B}_{\mathrm{KL}}(\widehat{\nu}, \varepsilon) \\ \mathrm{supp}(\nu) \subseteq (\widehat{\mathcal{S}} \cup \{x\})}} \nu(x), \tag{A.1}$$

where $\widehat{\mathcal{S}} = \mathrm{supp}(\widehat{\nu})$. To establish (A.1), it suffices to show that for any $\bar{\nu} \in \mathbb{B}_{\mathrm{KL}}(\widehat{\nu}, \varepsilon)$ that assigns a non-zero probability on $\mathcal{X} \backslash (\widehat{\mathcal{S}} \cup \{x\})$, there exists $\nu' \in \mathbb{B}_{\mathrm{KL}}(\widehat{\nu}, \varepsilon)$ satisfying $\mathrm{supp}(\nu') \subseteq \widehat{\mathcal{S}} \cup \{x\}$ such that $\nu'$ attains a higher objective value than $\bar{\nu}$, that is, $\nu'(x) > \bar{\nu}(x)$. Because $\bar{\nu}$ assigns a non-zero probability to $\mathcal{X} \backslash (\widehat{\mathcal{S}} \cup \{x\})$, we have

$$0 < \kappa \triangleq \sum_{z \in \mathcal{X} \backslash (\widehat{\mathcal{S}} \cup \{x\})} \bar{\nu}(z) \leq 1.$$

We now construct the measure $\nu'$ explicitly. Assume that $x \notin \widehat{\mathcal{S}}$. In this case, consider the discrete measure $\nu'$ supported on $\widehat{\mathcal{S}} \cup \{x\}$ given by

$$\nu'(x) = \bar{\nu}(x) + \kappa \quad \text{and} \quad \nu'(\widehat{x}_j) = \bar{\nu}(\widehat{x}_j) \quad \forall j \in [N].$$

Intuitively, $\nu'$ keeps the probability of $\bar{\nu}$ on $\widehat{\mathcal{S}}$, and it gathers the probability everywhere else and puts that mass onto $x$. We first show that $\nu'$ is a probability measure. Indeed, since $\kappa > 0$ and $\bar{\nu}$ is a probability measure, we have $\nu' \geq 0$. Moreover, we find

$$\sum_{z \in \mathcal{X}} \nu'(z) = \sum_{j \in [N]} \bar{\nu}(\widehat{x}_j) + \bar{\nu}(x) + \kappa = \sum_{j \in [N]} \bar{\nu}(\widehat{x}_j) + \bar{\nu}(x) + \sum_{z \in \mathcal{X} \backslash (\widehat{\mathcal{S}} \cup \{x\})} \bar{\nu}(z) = \sum_{z \in \mathcal{X}} \bar{\nu}(z) = 1,$$

where the first equality exploits the definition of $\bar{\nu}$, and the second equality follows from the definition of $\kappa$. Thus we conclude that $\nu'$ is a probability measure. We now proceed to show that $\nu'$ satisfies the KL divergence constraint. Indeed, we have

$$\mathrm{KL}(\widehat{\nu} \parallel \nu') = \sum_{z \in \mathcal{X}} f\left(\frac{\widehat{\nu}(z)}{\nu'(z)}\right) \nu'(z)$$

$$= \sum_{j \in [N]} f\left(\frac{\widehat{\nu}_j}{\nu'(\widehat{x}_j)}\right) \nu'(\widehat{x}_j) + \nu'(x) \tag{A.2a}$$

$$= \sum_{j \in [N]} f\left(\frac{\widehat{\nu}_j}{\bar{\nu}(\widehat{x}_j)}\right) \bar{\nu}(\widehat{x}_j) + \bar{\nu}(x) + \kappa \tag{A.2b}$$

$$= \sum_{j \in [N]} f\left(\frac{\widehat{\nu}_j}{\bar{\nu}(\widehat{x}_j)}\right) \bar{\nu}(\widehat{x}_j) + \bar{\nu}(x) + \sum_{z \in \mathcal{X} \backslash (\widehat{\mathcal{S}} \cup \{x\})} f\left(\frac{\widehat{\nu}(z)}{\bar{\nu}(z)}\right) \bar{\nu}(z) \tag{A.2c}$$

$$= \sum_{z \in \mathcal{X}} f\left(\frac{\widehat{\nu}(z)}{\bar{\nu}(z)}\right) \bar{\nu}(z) \leq \varepsilon. \tag{A.2d}$$

Equality (A.2a) holds because $f(0) = 1$ for the function $f$ defined in Definition 2.1 and $\mathrm{supp}(\nu') \subseteq \widehat{\mathcal{S}} \cup \{x\}$. Equality (A.2b) follows from the construction of $\nu'$, and equality (A.2c) holds due to the definition of $\kappa$ and the fact that $f(0) = 1$. Finally, the inequality in (A.2d) follows from the feasibility of $\bar{\nu}$, and it implies that $\nu' \in \mathbb{B}_{\mathrm{KL}}(\widehat{\nu}, \varepsilon)$. Furthermore, because $\kappa > 0$, we have $\nu'(x) = \bar{\nu}(x) + \kappa > \bar{\nu}(x)$ which asserts that $\bar{\nu}$ is strongly dominated by $\nu'$, and thus $\bar{\nu}$ cannot be an optimal measure.

Consider now the case $x \in \widehat{\mathcal{S}}$. Without loss of generality, we assume that $x = \widehat{x}_N$. In this case, it suffices to consider $\bar{\nu}$ satisfying $\bar{\nu}(\widehat{x}_N) \geq \widehat{\nu}_N$ because any $\bar{\nu}$ with $\bar{\nu}(\widehat{x}_N) < \widehat{\nu}_N$ is already dominated by the nominal measure $\widehat{\nu}$. Since $\kappa > 0$ and $\bar{\nu}(\widehat{x}_N) \geq \widehat{\nu}_N$, there must exist $K \in [N-1]$ atoms denoted without loss of generality by $\{\widehat{x}_1, \ldots, \widehat{x}_K\}$ that satisfy $\bar{\nu}(\widehat{x}_j) < \widehat{\nu}_j$ for all $k \in [K]$. Due to the continuity of the function $f$, there exists $\bar{\epsilon} \in (0, \kappa)$ that satisfies

$$f\left(\frac{\widehat{\nu}_N}{\bar{\nu}(\widehat{x}_N) + \bar{\epsilon}}\right)(\bar{\nu}(\widehat{x}_N) + \bar{\epsilon}) \leq f\left(\frac{\widehat{\nu}_N}{\bar{\nu}(\widehat{x}_N)}\right)\bar{\nu}(\widehat{x}_N) + \kappa.$$

We now consider the following measure $\nu'$ supported on $\widehat{\mathcal{S}}$:

$$\nu'(\widehat{x}_j) = \begin{cases} \bar{\nu}(\widehat{x}_j) + (\kappa - \bar{\epsilon}) \times (\widehat{\nu}_j - \bar{\nu}(\widehat{x}_j))/\sum_{k \in [K]}(\widehat{\nu}_k - \bar{\nu}(\widehat{x}_k)) & \forall j \in [K], \\ \bar{\nu}(\widehat{x}_j) & \forall j \in ([N-1]\setminus[K]), \\ \bar{\nu}(\widehat{x}_N) + \bar{\epsilon} & j = N. \end{cases}$$

We can verify that $\nu'$ is a probability measure supported on $\widehat{\mathcal{S}}$ and that $\nu'(\widehat{x}_N) > \bar{\nu}(\widehat{x}_N)$. Furthermore, we have

$$\mathrm{KL}(\widehat{\nu} \parallel \nu') = \sum_{j \in [N]} f\left(\frac{\widehat{\nu}_j}{\nu'(\widehat{x}_j)}\right)\nu'(\widehat{x}_j)$$

$$= \sum_{j \in [K]} f\left(\frac{\widehat{\nu}_j}{\nu'(\widehat{x}_j)}\right)\nu'(\widehat{x}_j) + \sum_{j \in ([N-1]\setminus[K])} f\left(\frac{\widehat{\nu}_j}{\nu'(\widehat{x}_j)}\right)\nu'(\widehat{x}_j) + f\left(\frac{\widehat{\nu}_N}{\nu'(\widehat{x}_N)}\right)\nu'(\widehat{x}_N)$$

$$\leq \sum_{j \in [N]} f\left(\frac{\widehat{\nu}_j}{\bar{\nu}(\widehat{x}_j)}\right)\bar{\nu}(\widehat{x}_j) + \kappa = \mathrm{KL}(\widehat{\nu} \parallel \bar{\nu}) \leq \varepsilon,$$

where the first inequality follows from the definition of $\nu'$, the definition of $\bar{\epsilon}$, the fact that for any $\widehat{\nu}_j > 0$ the function $t \mapsto tf(\widehat{\nu}_j/t)$ is non-increasing in $t$ over the domain $(0, \widehat{\nu}_j)$ and that $0 \leq \bar{\nu}(\widehat{x}_j) < \nu'(\widehat{x}_j) \leq \widehat{\nu}_j$ by construction. We have thus asserted that $\bar{\nu}$ is dominated by $\nu' \in \mathbb{B}_{\mathrm{KL}}(\widehat{\nu}, \varepsilon)$, and we conclude that (A.1) holds.

We now consider the supremum on the right hand side of (A.1). By Lemma A.1, the objective function of (A.1) is upper semicontinuous. Furthermore, the feasible set

$$\left\{\nu \in \mathcal{M}(\mathcal{X}) : \mathrm{supp}(\nu) \subseteq (\widehat{\mathcal{S}} \cup \{x\}),\ \mathrm{KL}(\widehat{\nu} \parallel \nu) \leq \varepsilon\right\}$$

is weakly compact because it only contains measures supported on a finite set [1, Theorem 15.11]. By the Weierstrass maximum value theorem [1, Theorem 2.43], the supremum in (A.1) is attained and there exists $\nu^\star_{\mathrm{KL}} \in \mathbb{B}_{\mathrm{KL}}(\widehat{\nu}, \varepsilon)$ such that

$$\sup_{\nu \in \mathbb{B}_{\mathrm{KL}}(\widehat{\nu}, \varepsilon)} \nu(x) = \nu^\star_{\mathrm{KL}}(x).$$

This observation completes the proof. $\qquad\square$

*Proof of Theorem 2.3.* Consider first the case when $x \in \widehat{\mathcal{S}}$, where $\widehat{\mathcal{S}} = \mathrm{supp}(\widehat{\nu})$. As a result of Proposition 2.2, the distribution that maximizes the probability at point $x$ subject to the KL divergence constraint will be supported on at most $N$ points from the set $\widehat{\mathcal{S}}$. The probability measures of interest thus share the form

$$\nu = \sum_{j \in [N]} y_j \delta_{\widehat{x}_j}$$

for some $y \in \mathbb{R}^N_+$, $\sum_{j \in [N]} y_j = 1$. The optimistic likelikood (5) satisfies

$$\nu^\star_{\mathrm{KL}}(x) = \sup\left\{\sum_{j \in [N]} y_j \mathbb{1}_x(\widehat{x}_j) : y \in \mathbb{R}^N_{++},\ \sum_{j \in [N]} \widehat{\nu}_j \log\left(\frac{\widehat{\nu}_j}{y_j}\right) \leq \varepsilon,\ \sum_{j \in [N]} y_j = 1\right\}, \qquad (\text{A.3})$$

which is a finite dimensional convex program in $y$.

Next, we consider the case where $x \notin \widehat{\mathcal{S}}$. To this end, for any $N \in \mathbb{N}_+$, we denote by $\Delta_N$ the simplex

$$\Delta_N \triangleq \left\{ y \in \mathbb{R}_+^N : 0 \leq y_j \leq 1 \, \forall j \in [N], \; \sum_{j \in [N]} y_j \leq 1 \right\}. \tag{A.4}$$

The relevant measures in $\mathbb{B}_{\mathrm{KL}}(\widehat{\nu}, \varepsilon)$ then share the form

$$\nu = \sum_{j \in [N]} y_j \delta_{\widehat{x}_j} + (1 - \sum_{j \in [N]} y_j) \delta_x$$

for some $y \in \Delta_N$. In this case, the optimistic likelihood 5 evaluates to

$$\nu_{\mathrm{KL}}^\star(x) = \max_{\substack{y \in \Delta_N \\ y > 0}} \left\{ 1 - \sum_{j \in [N]} y_j : \sum_{j \in [N]} y_j f \left( \frac{\widehat{\nu}_j}{y_j} \right) - \left( 1 - \sum_{j \in [N]} y_j \right) f(0) \leq \varepsilon \right\}.$$

Since $f$ is convex, the above program is a finite convex program in $y$. We now show that the above optimization problem admits an analytical solution. Consider the equivalent minimization problem

$$\mathrm{OPT}_{\mathrm{KL}}^\star \triangleq \min_{\substack{y \in \Delta_N \\ y > 0}} \left\{ \sum_{j \in [N]} y_j : \sum_{j \in [N]} \widehat{\nu}_j \log \widehat{\nu}_j - \sum_{j \in [N]} \widehat{\nu}_j \log y_j \leq \varepsilon \right\}. \tag{A.5}$$

Suppose that $\varepsilon > 0$. By a standard duality argument, the above program is equivalent to

$$\mathrm{OPT}_{\mathrm{KL}}^\star = \inf_{\substack{y \in \Delta_N \\ y > 0}} \sup_{\gamma \geq 0} \left\{ \sum_{j \in [N]} y_j + \gamma \left( \sum_{j \in [N]} \widehat{\nu}_j \log \widehat{\nu}_j - \varepsilon - \sum_{j \in [N]} \widehat{\nu}_j \log y_j \right) \right\} \tag{A.6a}$$

$$= \sup_{\gamma \geq 0} \left\{ \gamma \left( \sum_{j \in [N]} \widehat{\nu}_j \log \widehat{\nu}_j - \varepsilon \right) + \inf_{\substack{y \in \Delta_N \\ y > 0}} \left\{ \sum_{j \in [N]} y_j - \gamma \sum_{j \in [N]} \widehat{\nu}_j \log y_j \right\} \right\} \tag{A.6b}$$

$$\geq \sup_{1 \geq \gamma > 0} \left\{ \gamma \left( \sum_{j \in [N]} \widehat{\nu}_j \log \widehat{\nu}_j - \varepsilon \right) + \inf_{\substack{y \in \Delta_N \\ y > 0}} \left\{ \sum_{j \in [N]} y_j - \gamma \sum_{j \in [N]} \widehat{\nu}_j \log y_j \right\} \right\} \tag{A.6c}$$

$$= \sup_{1 \geq \gamma > 0} \left\{ \gamma \left( \sum_{j \in [N]} \widehat{\nu}_j - \varepsilon \right) - \sum_{j \in [N]} \widehat{\nu}_j \gamma \log \gamma \right\}, \tag{A.6d}$$

where the equality (A.6b) follows from strong duality since the Slater condition for the primal problem is satisfied. The inequality (A.6c) follows directly from the restriction of the feasible set of $\gamma$ and because the objective function is continuous in $\gamma$. For any $\gamma \in (0, 1]$, the inner minimization admits the optimal solution $y_j^\star = \gamma \widehat{\nu}_j$, and this leads to the last equation (A.6d). The maximization over $\gamma$ is now a convex optimization problem, and the first-order condition gives the optimal solution $\gamma^\star = \exp(-\varepsilon)$. We can thus conclude that

$$\mathrm{OPT}_{\mathrm{KL}}^\star \geq \exp(-\varepsilon).$$

By substituting the feasible solution

$$y_j = \exp(-\varepsilon) \widehat{\nu}_j \quad \forall j \in [N]$$

into (A.6a), we see that $\mathrm{OPT}_{\mathrm{KL}}^\star \leq \exp(-\varepsilon)$. Hence,

$$\mathrm{OPT}_{\mathrm{KL}}^\star = \exp(-\varepsilon) \quad \forall \varepsilon > 0.$$

Consider now the optimal value $\mathrm{OPT}_{\mathrm{KL}}^\star$ defined in (A.5) as a parametric function of the radius $\varepsilon$ over the domain $\mathbb{R}_+$. One can show that $\mathrm{OPT}_{\mathrm{KL}}^\star$ is a continuous function over $\varepsilon \in \mathbb{R}_+$ using Berge's maximum theorem [1, Theorem 17.31]. Furthermore, the function $\exp(-\varepsilon)$ is also continuous over $\varepsilon \in \mathbb{R}_+$. We thus conclude that

$$\mathrm{OPT}_{\mathrm{KL}}^\star = \exp(-\varepsilon) \quad \forall \varepsilon \geq 0.$$

The proof for this case is completed by noticing that $\nu_{\mathrm{KL}}^\star(x) = 1 - \mathrm{OPT}_{\mathrm{KL}}^\star$. $\qquad \square$

## A.2 Proofs of Section 4

*Proof of Proposition 4.2.* When $\varepsilon = 0$, $\mathbb{B}_{\mathrm{W}}(\widehat{\nu}, \varepsilon)$ is the singleton set $\{\widehat{\nu}\}$ and the claim is trivial. It thus suffices to consider $\varepsilon > 0$. Since $\mathbb{B}_{\mathrm{W}}(\widehat{\nu}, \varepsilon)$ is weakly compact [8, Proposition 3] and the objective function in (8) is upper-semicontinuous in $\nu$ by Lemma A.1, a version of the Weierstrass maximum value theorem [1, Theorem 2.43] implies that there exists $\nu^{\star} \in \mathbb{B}_{\mathrm{W}}(\widehat{\nu}, \varepsilon)$ such that

$$\sup_{\nu \in \mathbb{B}_{\mathrm{W}}(\widehat{\nu}, \varepsilon)} \nu(x) = \nu_{\mathrm{W}}^{\star}(x).$$

Suppose that $\bar{\nu}$ is an optimal measure that solves (8), that is, $\bar{\nu} \in \mathbb{B}_{\mathrm{W}}(\widehat{\nu}, \varepsilon)$ and $\bar{\nu}(x) = \nu_{\mathrm{W}}^{\star}(x)$. Since the ground metric distance $d(\cdot, \cdot)$ in the Wasserstein distance is continuous, there exists an optimal transport plan $\bar{\lambda}$ that maps $\widehat{\nu}$ to $\bar{\nu}$ [10, Theorem 4.1]. Since $\widehat{\nu}$ is a discrete distribution with $N$ atoms, this optimal transport map can be characterized by $N$ functions $\bar{\lambda}_j : \mathcal{X} \to \mathbb{R}_+$, $j \in [N]$, which satisfy the balancing constraints

$$\sum_{z \in \mathcal{X}} \bar{\lambda}_j(z) = \widehat{\nu}_j \; \forall j \in [N] \quad \text{and} \quad \sum_{j=1}^{N} \bar{\lambda}_j(z) = \bar{\nu}(z) \; \forall z \in \mathcal{X}$$

as well as the Wasserstein distance constraint

$$\sum_{j \in [N]} \sum_{z \in \mathcal{X}} d(\widehat{x}_j, z) \bar{\lambda}_j(z) \le \varepsilon. \tag{A.7}$$

Define $\kappa_j$ and $\eta_j$ as

$$\kappa_j \triangleq \sum_{z \in \mathcal{X} \backslash (\widehat{\mathcal{S}} \cup \{x\})} \bar{\lambda}_j(z) \quad \text{and} \quad \eta_j \triangleq \sum_{z \in \mathcal{X} \backslash (\widehat{\mathcal{S}} \cup \{x\})} d(\widehat{x}_j, z) \bar{\lambda}_j(z) \qquad \forall j \in [N].$$

By construction, we have $0 \le \kappa_j \le \widehat{\nu}_j \le 1$ and $0 \le \eta_j$ for all $j \in [N]$. Suppose that $\bar{\nu}$ assigns non-zero probability mass on $\mathcal{X} \backslash (\widehat{\mathcal{S}} \cup \{x\})$, where $\widehat{\mathcal{S}} = \mathrm{supp}(\widehat{\nu})$. In that case, there exists $j \in [N]$ such that $\kappa_j > 0$ and $\eta_j > 0$. We will next show that $\bar{\nu}$ cannot be the optimal solution.

Assume first that $x \notin \widehat{\mathcal{S}}$, and define the transport maps $\lambda'_j : \mathcal{X} \to \mathbb{R}_+$ for $j \in [N]$ as

$$\lambda'_j(z) = \begin{cases} \bar{\lambda}_j(\widehat{x}_j) + \left(1 - \min\left\{1, \frac{\eta_j}{d(x, \widehat{x}_j)}\right\}\right) \kappa_j & \text{if } z = \widehat{x}_j, \\ \bar{\lambda}_j(\widehat{x}_k) & \text{if } z = \widehat{x}_k, k \neq j, k \in [N], \\ \bar{\lambda}_j(x) + \min\left\{1, \frac{\eta_j}{d(x, \widehat{x}_j)}\right\} \kappa_j & \text{if } z = x, \\ 0 & \text{otherwise.} \end{cases}$$

By this construction of $\lambda'_j$, we obtain

$$\sum_{z \in \mathcal{X}} \lambda'_j(z) = \sum_{z \in \mathcal{X}} \bar{\lambda}_j(z) = \widehat{\nu}_j \qquad \forall j \in [N].$$

We now construct a measure $\nu'$ explicitly using the transport map $\lambda'$ from $\widehat{\nu}$ as

$$\nu'(z) = \sum_{j \in [N]} \lambda'_j(z) \qquad \forall z \in \mathcal{X}. \tag{A.8}$$

Notice that $\nu'$ is supported on $\widehat{\mathcal{S}} \cup \{x\}$, $\nu' \ge 0$ and

$$\sum_{z \in \mathcal{X}} \nu'(z) = \sum_{j \in [N]} \left( \sum_{k \in [N]} \bar{\lambda}_j(\widehat{x}_k) + \kappa_j + \bar{\lambda}_j(x) \right) = \sum_{j \in [N]} \sum_{z \in \mathcal{X}} \bar{\lambda}_j(z) = \sum_{j \in [N]} \widehat{\nu}_j = 1,$$

which further implies that $\nu'$ is a probability measure on $\mathcal{X}$. Moreover, we have

$$\mathbb{W}(\widehat{\nu}, \nu') \leq \sum_{j \in [N]} \sum_{k \in [N]} d(\widehat{x}_j, \widehat{x}_k) \lambda'_j(\widehat{x}_k) + \sum_{j \in [N]} d(\widehat{x}_j, x) \lambda'_j(x) \tag{A.9a}$$

$$= \sum_{j \in [N]} \left( \sum_{k \in [N]} d(\widehat{x}_j, \widehat{x}_k) \bar{\lambda}_j(\widehat{x}_k) + d(\widehat{x}_j, x) \bar{\lambda}_j(x) + \min \left\{ d(\widehat{x}_j, x) \kappa_j, \eta_j \kappa_j \right\} \right)$$

$$\leq \sum_{j \in [N]} \left( \sum_{k \in [N]} d(\widehat{x}_j, \widehat{x}_k) \bar{\lambda}_j(\widehat{x}_k) + d(\widehat{x}_j, x) \bar{\lambda}_j(x) + \eta_j \kappa_j \right)$$

$$\leq \sum_{j \in [N]} \left( \sum_{k \in [N]} d(\widehat{x}_j, \widehat{x}_k) \bar{\lambda}_j(\widehat{x}_k) + d(\widehat{x}_j, x) \bar{\lambda}_j(x) + \eta_j \right) \tag{A.9b}$$

$$= \sum_{j \in [N]} \left( \sum_{k \in [N]} d(\widehat{x}_j, \widehat{x}_k) \bar{\lambda}_j(\widehat{x}_k) + d(\widehat{x}_j, x) \bar{\lambda}_j(x) + \sum_{z \in \mathcal{X} \setminus (\widehat{\mathcal{S}} \cup \{x\})} d(\widehat{x}_j, z) \bar{\lambda}_j(z) \right)$$

$$= \sum_{j \in [N]} \sum_{z \in \mathcal{X}} d(\widehat{x}_j, z) \bar{\lambda}_j(z) \leq \varepsilon. \tag{A.9c}$$

Inequality (A.9a) holds because of the definition of the Wasserstein distance and the fact that $\{\lambda'_j\}_{j \in [N]}$ constitutes a feasible transportation plan from $\widehat{\nu}$ to $\nu'$. Inequality (A.9b) holds due to the non-negativity of both $\eta_j$ and $\kappa_j$ and the fact that $\kappa_j \leq 1$. Inequality (A.9c) is a consequence of (A.7). The last inequality implies that $\nu' \in \mathbb{B}_{\mathbb{W}}(\widehat{\nu}, \varepsilon)$, and thus $\nu'$ is a feasible measure for the optimistic likelihood problem. Finally, we have

$$\nu'(x) = \sum_{j \in [N]} \lambda'_j(x) = \sum_{j \in [N]} \left( \bar{\lambda}_j(x) + \min \left\{ 1, \frac{\eta_j}{d(x, \widehat{x}_j)} \right\} \kappa_j \right) > \sum_{j \in [N]} \bar{\lambda}_j(x) = \bar{\nu}(x),$$

where the strict inequality is from the fact that there exists $j \in [N]$ such that $\kappa_j > 0$ and $\eta_j > 0$. Thus, $\nu' \in \mathbb{B}_{\mathbb{W}}(\widehat{\nu}, \varepsilon)$ attains a higher objective value than $\bar{\nu}$, and as a consequence $\bar{\nu}$ cannot be an optimal measure. Notice that $\text{supp}(\nu') \subseteq (\widehat{\mathcal{S}} \cup \{x\})$ by construction, and thus we conclude that when $x \notin \widehat{\mathcal{S}}$, the optimal measure $\nu^\star_{\mathbb{W}}$ satisfies $\text{supp}(\nu^\star_{\mathbb{W}}) \subseteq (\widehat{\mathcal{S}} \cup \{x\})$.

Assume now that $x \in \widehat{\mathcal{S}}$, and assume without loss of generality that $x = \widehat{x}_N$. Consider now the transport plan $\lambda'_j : \mathcal{X} \to \mathbb{R}_+$ for any $j \in [N]$ defined as

$$\forall j \in [N-1]: \ \lambda'_j(z) = \begin{cases} \bar{\lambda}_j(\widehat{x}_j) + \left( 1 - \min \left\{ 1, \frac{\eta_j}{d(x, \widehat{x}_j)} \right\} \right) \kappa_j & \text{if } z = \widehat{x}_j, \\ \bar{\lambda}_j(\widehat{x}_k) & \text{if } z = \widehat{x}_k, k \neq j, k \in [N-1], \\ \bar{\lambda}_j(x) + \min \left\{ 1, \frac{\eta_j}{d(x, \widehat{x}_j)} \right\} \kappa_j & \text{if } z = \widehat{x}_N, \\ 0 & \text{otherwise} \end{cases}$$

and

$$\lambda'_N(z) = \begin{cases} \bar{\lambda}_N(\widehat{x}_k) & \text{if } z = \widehat{x}_k, k \in [N-1], \\ \bar{\lambda}_N(\widehat{x}'_N) + \kappa_N & \text{if } z = \widehat{x}_N, \\ 0 & \text{otherwise.} \end{cases}$$

One can readily verify that using the collection $\{\lambda'_j\}_{j \in [N]}$ to define $\nu'$ in (A.8) results in a probability measure $\nu' \in \mathbb{B}_{\mathbb{W}}(\widehat{\nu}, \varepsilon)$ that attains a strictly higher objective value than $\bar{\nu}$. Notice that this construction satisfies $\text{supp}(\nu') \subseteq \widehat{\mathcal{S}}$, and hence we can conclude that when $x \in \widehat{\mathcal{S}}$, the optimal measure $\nu^\star_{\mathbb{W}}$ satisfies $\text{supp}(\nu^\star_{\mathbb{W}}) \subseteq \widehat{\mathcal{S}}$. This completes the proof. $\qquad\square$

*Proof of Theorem 4.3.* As a result of Proposition 4.2, we can restrict ourselves to probability measures that are supported on $\text{supp}(\widehat{\nu}) \cup \{x\}$. Thus, it suffices to optimize over the set of discrete probability measures of the form

$$\nu = \sum_{j \in [N]} y_j \delta_{\widehat{x}_j} + \left( 1 - \sum_{j \in [N]} y_j \right) \delta_x$$

for some $y \in \Delta_N$, where $\Delta_N$ is the simplex defined in (A.4). Using the Definition 4.1 of the type-1 Wasserstein distance, we can rewrite the optimistic likelihood problem over the Wasserstein ball $\mathbb{B}_W(\widehat{\nu}, \varepsilon)$ as the linear program

$$
\sup_{\nu \in \mathbb{B}_W(\widehat{\nu},\varepsilon)} \nu(x) = \begin{cases} \sup & 1 - \sum_{j \in [N]} y_j \\ \text{s.t.} & y \in \Delta_N,\ \lambda \in \mathbb{R}_+^{N \times (N+1)} \\ & \sum_{j \in [N]} \sum_{j' \in [N]} d(\widehat{x}_j, \widehat{x}_{j'})\lambda_{jj'} + \sum_{j \in [N]} d(\widehat{x}_j, x)\lambda_{j(N+1)} \le \varepsilon \\ & \sum_{j' \in [N+1]} \lambda_{jj'} = \widehat{\nu}_j & \forall j \in [N] \\ & \sum_{j \in [N]} \lambda_{jj'} = y_j & \forall j' \in [N] \\ & \sum_{j \in [N]} \lambda_{j(N+1)} = 1 - \sum_{j \in [N]} y_j. \end{cases}
$$

From the last constraint, we can see that maximizing $1 - \sum_{j \in [N]} y_j$ is equivalent to maximizing $\sum_{j \in [N]} \lambda_{j(N+1)}$. In particular, we thus conclude that it is optimal to set $\lambda_{jj'} = 0$ for any $j \in [N], j' \in [N]$ such that $j \ne j'$. We thus have

$$
\sup_{\nu \in \mathbb{B}_W(\widehat{\nu},\varepsilon)} \nu(x) = \begin{cases} \sup & \sum_{j \in [N]} \lambda_{j(N+1)} \\ \text{s.t.} & y \in \Delta_N,\ \lambda \in \mathbb{R}_+^{N \times (N+1)} \\ & \lambda_{jj'} = 0 \quad \forall j \in [N], j' \in [N], j \ne j' \\ & \sum_{j \in [N]} d(\widehat{x}_j, x)\,\lambda_{j(N+1)} \le \varepsilon \\ & \lambda_{jj} + \lambda_{j(N+1)} = \widehat{\nu}_j, \quad \lambda_{jj} = y_j \qquad \forall j \in [N]. \end{cases}
$$

By letting $T_j = \lambda_{j(N+1)}$ and eliminating the redundant components of $\lambda$, we obtain the desired reformulation. This completes the proof. $\qquad\square$

*Proof of Proposition 4.4.* By a change of variables, we define the weight $\widehat{w}_j = d(\widehat{x}_j, x)\widehat{\nu}_j$ and the decision variables $z_j = \widehat{\nu}_j^{-1} T_j$ for every $j \in [N]$. The optimal value of problem (9) then coincides with the optimal value of

$$
\max \left\{ \sum_{j \in [N]} \widehat{\nu}_j z_j : z \in \mathbb{R}_+^N,\ \sum_{j \in [N]} \widehat{w}_j z_j \le \varepsilon,\ z_j \le 1 \ \forall j \in [N] \right\}, \tag{A.10}
$$

which is a continuous (or fractional) knapsack problem. The special structure of (A.10) guarantees

$$
\frac{\widehat{\nu}_j}{\widehat{w}_j} = \frac{1}{d(\widehat{x}_j, x)} \quad \forall j \in [N],
$$

and hence the continuous knapsack problem (A.10) admits an optimal solution $z^\star$ that can be found by sorting the support points $\widehat{x}_j$ in increasing order of distance from $x$ and then exhausting the budget $\varepsilon$ according to the sorted order (see [3] or [6, Proposition 17.1]). Since sorting an array of $N$ scalars can be achieved in time $\mathcal{O}(N \log N)$, problem (A.10) can be solved efficiently, and the optimal solution $T^\star$ of (9) can be constructed from the optimal solution $z^\star$ of (A.10) by setting

$$
T_j^\star = \widehat{\nu}_j z_j^\star \quad \forall j \in [N].
$$

This completes the proof. $\qquad\square$

**Corollary A.2** (Comparative statics). *If the radius $\varepsilon$ of the Wasserstein ball is strictly positive, then $\nu_W^\star(x) > 0$. Moreover, if the radius satisfies $\varepsilon \ge \sum_{j \in [N]} d(x, \widehat{x}_j)\widehat{\nu}_j$, then $\nu_W^\star(x) = 1$.*

The proof of Corollary A.2 follows directly from examining the optimal value of the linear program (9) and is thus omitted.

## A.3 Proofs of Section 5

In the proofs of this section, we denote by $\nu_i^{\text{true}}$ the unknown true probability measure that induces the probability mass function $p(\cdot|\theta_i)$ for each $i \in [C]$.

*Proof of Theorem 5.3.* Define for each $i \in [C]$ the set

$$\Phi_i \triangleq \left\{ \nu_i \in \mathcal{M}(\mathcal{X}) : \text{KL}(\nu_i \parallel \nu_i^{\text{true}}) > \varepsilon_i \right\},$$

where the dependence of $\Phi_i$ on $\varepsilon_i$ and $\nu_i^{\text{true}}$ has been made implicit. Under Assumption 5.2, the empirical measure $\widehat{\nu}_i^{N_i}$ satisfies the large deviation principle with rate function $\text{KL}(\cdot \parallel \nu_i^{\text{true}})$ [4, Theorem 6.2.10]. Sanov's theorem then implies that

$$\limsup_{N_i \to \infty} \frac{1}{N_i} \log \mathbb{P}^{\infty}\left(\widehat{\nu}_i^{N_i} \in \Phi_i\right) \leq -\varepsilon_i < 0 \qquad \forall i \in [C]. \tag{A.11}$$

This in turn implies that there exist positive constants $\kappa_i < \infty$ such that

$$\mathbb{P}^{N_i}\left(\widehat{\nu}_i^{N_i} \in \Phi_i\right) \leq \kappa_i \exp(-N_i \varepsilon_i) \quad \text{as } N_i \to \infty.$$

We now have

$$\mathbb{P}^{\infty}(\mathcal{J}^{\text{true}} \geq \widehat{\mathcal{J}}_{\mathbb{B}^N}) \geq \mathbb{P}^{\infty}\left(\nu_i^{\text{true}} \in \mathbb{B}_{\text{KL}}(\widehat{\nu}_i^{N_i}, \varepsilon_i) \, \forall i \in [C]\right) \tag{A.12}$$

$$= \prod_{i \in [C]} \mathbb{P}^{N_i}\left(\nu_i^{\text{true}} \in \mathbb{B}_{\text{KL}}(\widehat{\nu}_i^{N_i}, \varepsilon_i)\right) \tag{A.13}$$

$$= \prod_{i \in [C]} \left(1 - \mathbb{P}^{N_i}\left(\widehat{\nu}_i^{N_i} \in \Phi_i\right)\right) \tag{A.14}$$

$$\geq 1 - \sum_{i \in [C]} \mathbb{P}^{N_i}\left(\widehat{\nu}_i^{N_i} \in \Phi_i\right). \tag{A.15}$$

Here, equality (A.13) follows from our i.i.d. assumption. Equality (A.14) follows from the fact that the event $\nu_i^{\text{true}} \in \mathbb{B}_{\text{KL}}(\widehat{\nu}_i^{N_i}, \varepsilon_i)$ is the complement of the event $\widehat{\nu}_i^{N_i} \in \Phi_i$. Inequality (A.15), finally, is due to the Weierstrass product inequality. Thus, for each $i$ there exists $C_i < \infty$ such that as $N_i \to \infty$, we have

$$\mathbb{P}^{\infty}(\mathcal{J}^{\text{true}} < \widehat{\mathcal{J}}_{\mathbb{B}^N}) \leq \sum_{i \in [C]} \mathbb{P}^{N_i}\left(\widehat{\nu}_i^{N_i} \in \Phi_i\right) \leq \sum_{i \in [C]} \kappa_i \exp\left(-N_i \varepsilon_i\right) \leq \kappa C \exp\left(-n \min_{i \in [C]}\{\varepsilon_i\}\right)$$

for some $\kappa = \max_{i \in [C]} \kappa_i < \infty$. This further implies that

$$\limsup_{n \to \infty} \frac{1}{n} \log \mathbb{P}^{\infty}(\mathcal{J}^{\text{true}} < \widehat{\mathcal{J}}_{\mathbb{B}^N}) \leq - \min_{i \in [C]}\{\varepsilon_i\} < 0.$$

This observation completes the proof. $\qquad \square$

*Proof of Theorem 5.5.* If $\varepsilon_i$ is chosen as in the statement of the theorem, then the measure concentration result for the Wasserstein distance [5, Theorem 2] implies that

$$\mathbb{P}^{N_i}\left(\mathbb{W}(\nu_i^{\text{true}}, \widehat{\nu}_i^{N_i}) \geq \varepsilon_i(\beta, C, N_i)\right) \leq \frac{\beta}{C}.$$

Thus, by applying the union bound, we obtain

$$\mathbb{P}^{N}\left(\mathbb{W}(\nu_i^{\text{true}}, \widehat{\nu}_i^{N_i}) \geq \varepsilon_i(\beta, C, N_i) \, \forall i\right) = \sum_i \mathbb{P}^{N_i}\left(\mathbb{W}(\nu_i^{\text{true}}, \widehat{\nu}_i^{N_i}) \geq \varepsilon_i(\beta, C, N_i)\right) \leq \beta,$$

which implies that

$$\mathbb{P}^{N}\left(\nu_i^{\text{true}} \in \mathbb{B}_{\mathbb{W}}\left(\widehat{\nu}_i^{N_i}, \varepsilon_i(\beta, C, N_i)\right) \, \forall i\right) \geq 1 - \beta.$$

We can now conclude that $\widehat{\mathcal{J}}_{\mathbb{B}^N} \leq \mathcal{J}^{\text{true}}$ with probability at least $1 - \beta$. $\qquad \square$

*Proof of Theorem 5.6.* For every $i \in [C]$, let $\nu_i^\star \in \mathbb{B}_i^{N_i}(\widehat{\nu}_i^{N_i})$ be an optimal solution of the problem

$$\sup_{\nu_i \in \mathbb{B}_i^{N_i}(\widehat{\nu}_i^{N_i})} \nu_i(x), \tag{A.16}$$

where the dependence of $\nu_i^\star$ on the number of samples $N_i$ has been omitted to avoid clutter. The existence of $\nu_i^\star \in \mathbb{B}_i^{N_i}(\widehat{\nu}_i^{N_i})$ is guaranteed by Proposition 4.2. By [7, Lemma 3.7], for every $i \in [C]$ it holds $(\nu_i^{\text{true}})^\infty$-almost surely that

$$\lim_{N_i \to \infty} \mathbb{W}\left(\nu_i^{\text{true}}, \nu_i^\star\right) = 0.$$

Therefore, by [10, Theorem 6.9], $\nu_i^\star$ converges to $\nu_i^{\text{true}}$ weakly as $N_i \to \infty$. Since $\mathbb{1}_x(\cdot)$ is a bounded, upper semicontinuous function, the weak continuity implies that $(\nu_i^{\text{true}})^\infty$-almost surely as $N_i \to \infty$, we have that

$$\nu_i^\star(x) \to \nu_i^{\text{true}}(x) = p(x|\theta_i). \tag{A.17}$$

Let $u^{\text{true}} \in [0,1]^C$ be the vector defined by $(u^{\text{true}})_i = p(x|\theta_i)$ for $i \in [C]$. Since $(u^{\text{true}})_i > 0$ for $i = 1, \ldots, C$, there exists $\underline{u} > 0$ such that $u^{\text{true}} \in [\underline{u}, 1]^C$. Consider the parametrized optimization problems

$$\mathcal{J}^\star(u) \triangleq \min_{q \in \mathcal{Q}} \left\{ \mathcal{J}(q,u) \triangleq \sum_{i \in [C]} q_i(\log q_i - \log \pi_i) - \sum_{i \in [C]} q_i \log u_i \right\}, \quad u \in [\underline{u}, 1]^C.$$

We observe that $\mathcal{J}(\cdot, \cdot)$ is jointly continuous on $\mathcal{Q} \times [\underline{u}, 1]^C$, $\mathcal{Q}$ is compact, and the level sets

$$\left\{ q \in \mathcal{Q} : \mathcal{J}(q,u) \leq -\sum_{i \in [C]} \pi_i \log \underline{u} \right\}$$

are non-empty and uniformly bounded over all $u \in [\underline{u}, 1]^C$. By [2, Proposition 4.4] and the discussion following its proof, $\mathcal{J}^\star(u)$ is continuous on $[\underline{u}, 1]^C$. The continuity of $\mathcal{J}^\star(\cdot)$ and the convergence (A.17) together imply that $(\nu_1^{\text{true}})^\infty \times \cdots \times (\nu_C^{\text{true}})^\infty$-almost surely, and we thus have

$$\widehat{\mathcal{J}}_{\mathbb{B}^N} = \mathcal{J}^\star((\nu_1^\star(x), \ldots, \nu_C^\star(x))) \to \mathcal{J}^\star(u^{\text{true}}) = \mathcal{J}^{\text{true}} \quad \text{as } N_1, \ldots, N_C \to \infty.$$

This observation completes the proof. $\qquad\square$

# Appendix B    Additional Material

## B.1    A Measure-Theoretic Derivation of the Evidence Lower Bound Problem

To keep the paper self-contained, we present in this section a derivation of the evidence lower bound (ELBO), which is a fundamental building block of the variational Bayes method.

Consider a standard Bayesian inference model where the random vector $\tilde{x}$, supported on a sample space $\mathcal{X}$, is governed by one of the distributions $\mathbb{P}_\theta$ parameterized by $\theta \in \Theta$. We assume that there exists a measure $\bar{\nu}$ on $\mathcal{X}$ such that $\mathbb{P}_\theta$ is absolutely continuous with respect to $\bar{\nu}$ for all $\theta \in \Theta$. Moreover, we denote by $f_{\tilde{x}|\theta}$ the Radon-Nikodym derivative of $\mathbb{P}_\theta$ with respect to $\bar{\nu}$, that is

$$f_{\tilde{x}|\theta}(x|\theta) = \frac{\mathrm{d}\mathbb{P}_\theta}{\mathrm{d}\bar{\nu}}(x) \quad \forall x \in \mathcal{X}.$$

Finally, we denote by $\pi$ the prior measure on the parameter space $\Theta$, while $\mathbb{P}_x$ denotes the posterior measure on $\Theta$ after observing $x$.

Consider an optimal solution $\mathbb{Q}^\star$ of the optimization problem

$$\mathbb{Q}^\star \in \arg\min_{\mathbb{Q} \in \mathcal{Q}} \mathrm{KL}(\mathbb{Q} \parallel \mathbb{P}_x),$$

where $\mathrm{KL}(\cdot \parallel \cdot)$ denotes the KL divergence defined in Definition 2.1. If the feasible set $\mathcal{Q}$ is the collection of all possible probability measures supported on $\Theta$, then $\mathbb{Q}^\star = \mathbb{P}_x$. The objective function of this problem can be re-expressed as

$$\mathrm{KL}(\mathbb{Q} \parallel \mathbb{P}_x) = \int_\Theta \log\left(\frac{\mathrm{d}\mathbb{Q}}{\mathrm{d}\mathbb{P}_x}\right) \mathrm{d}\mathbb{Q} \tag{B.1a}$$

$$= \int_\Theta \log\left(\frac{\mathrm{d}\mathbb{Q}}{\mathrm{d}\pi}\right) \mathrm{d}\mathbb{Q} - \int_\Theta \log\left(\frac{\mathrm{d}\mathbb{P}_x}{\mathrm{d}\pi}\right) \mathrm{d}\mathbb{Q} \tag{B.1b}$$

$$= \mathrm{KL}(\mathbb{Q} \parallel \pi) - \int_\Theta \log\left(\frac{\mathrm{d}\mathbb{P}_\theta}{\mathrm{d}\bar{\nu}}(x)\right) \mathrm{d}\mathbb{Q} + \log \int_\Theta f_{\tilde{x}|\theta}(x|\theta)\mathrm{d}\pi, \tag{B.1c}$$

where the equality (B.1a) follows from the definition of KL divergence, and (B.1b) is due to the chain rule for the Radon-Nikodym derivatives because $\mathbb{P}_x \ll \pi$ [9, Theorem 1.31]. Equality (B.1c), finally, holds since

$$\frac{\mathrm{d}\mathbb{P}_x}{\mathrm{d}\pi}(\theta) = \frac{f_{\tilde{x}|\theta}(x|\theta)}{\int_\Theta f_{\tilde{x}|\theta}(x|\theta)\mathrm{d}\pi(\theta)} = \frac{1}{\int_\Theta f_{\tilde{x}|\theta}(x|\theta)\mathrm{d}\pi(\theta)} \cdot \frac{\mathrm{d}\mathbb{P}_\theta}{\mathrm{d}\bar{\nu}}(x),$$

where the first equality follows from Bayes' theorem [9, Theorem 1.31] and the second equality is due to the definition of $f_{\tilde{x}|\theta}$. Since the last term in (B.1c) does not involve the decision variable $\mathbb{Q}$, the measure $\mathbb{Q}^\star$ can be equivalently expressed as the optimal solution of

$$\min_{\mathbb{Q} \in \mathcal{Q}} \mathrm{KL}(\mathbb{Q} \parallel \pi) - \int_\Theta \log\left(\frac{\mathrm{d}\mathbb{P}_\theta}{\mathrm{d}\bar{\nu}}(x)\right) \mathrm{d}\mathbb{Q}.$$

If we define the conditional density $p(x|\theta)$ with respect to $\bar{\nu}$ of $\tilde{x}$ given the parameter $\theta$ [9, Section 1.3.1], that is,

$$p(x|\theta) = f_{\tilde{x}|\theta}(x|\theta),$$

then $\mathbb{Q}^\star$ solves

$$\min_{\mathbb{Q} \in \mathcal{Q}} \mathrm{KL}(\mathbb{Q} \parallel \pi) - \mathbb{E}_\mathbb{Q}[\log p(x|\theta)].$$

The function $p(x|\theta)$, considered as a function of the parameter $\theta$ after $x$ has been observed, is often called the *likelihood function*. If $p(x|\theta)$ is considered as a function of $x$ given the parameter $\theta$, then it is often called the *conditional density*.

## B.2    Generalization to $f$-Divergence Ambiguity Sets

In this section, we consider the class of ambiguity sets described by $f$-divergences, which generalizes the KL ambiguity set from Section 2.

**Definition B.1** ($f$-divergence). The $f$-divergence $D_f$ between two measures $\nu_1$ and $\nu_2$ supported on $\mathcal{X}$ is defined as

$$D_f(\nu_1 \| \nu_2) = \int_{z \in \mathcal{X}} f\left(\frac{\nu_1(z)}{\nu_2(z)}\right) \nu_2(z),$$

where $f : \mathbb{R} \to \mathbb{R}$ is a convex function satisfying $f(1) = 0$. More specifically,

- If $f(t) = t\log(t) - t + 1$, then $D_f$ is the *Kullback-Leibler divergence*.

- If $f(t) = 1 - \sqrt{t}$, then $D_f$ is the *Hellinger distance*.

- If $f(t) = (t-1)^2$, then $D_f$ is the *Pearson's $\chi^2$-divergence*.

- If $f(t) = |t - 1|$, then $D_f$ is the *total variation distance*.

We now consider the $f$-divergence ball $\mathbb{B}_f(\widehat{\nu}, \varepsilon)$ of radius $\varepsilon \geq 0$, which contains all probability measures in the neighborhood of $\widehat{\nu}$ as measured by the $f$-divergence:

$$\mathbb{B}_f(\widehat{\nu}, \varepsilon) \triangleq \{\nu \in \mathcal{M}(\mathcal{X}) : D_f(\widehat{\nu} \| \nu) \leq \varepsilon\} \tag{B.2}$$

Moreover, we assume that the nominal distribution $\widehat{\nu}$ is supported on $N$ distinct points $\widehat{x}_1, \ldots, \widehat{x}_N$, that is, $\widehat{\nu} = \sum_{j \in [N]} \widehat{\nu}_j \delta_{\widehat{x}_j}$ with $\widehat{\nu}_j > 0 \, \forall j \in [N]$ and $\sum_{j \in [N]} \widehat{\nu}_j = 1$.

In analogy to Section 2, we first provide a generalized version of Proposition 2.2.

**Corollary B.2** (Existence of optimizers; $f$-divergence ambiguity). For any $\varepsilon \geq 0$ and $x \in \mathcal{X}$, there exists a measure $\nu_f^\star \in \mathbb{B}_f(\widehat{\nu}, \varepsilon)$ such that

$$\sup_{\nu \in \mathbb{B}_f(\widehat{\nu}, \varepsilon)} \nu(x) = \nu_f^\star(x). \tag{B.3}$$

Moreover, $\nu_f^\star$ is supported on at most $N + 1$ points satisfying $\mathrm{supp}(\nu_f^\star) \subseteq \mathrm{supp}(\widehat{\nu}) \cup \{x\}$.

The proof of Corollary B.2 follows from the proof of Proposition 2.2 and thus it is omitted.

**Theorem B.3** (Optimistic likelihood; $f$-divergence ambiguity). Suppose that $\widehat{\nu} = \sum_{j \in [N]} \widehat{\nu}_j \delta_{\widehat{x}_j}$. For any data point $x \in \mathcal{X}$, the optimization problem in (B.3) can be reformulated as a finite convex program. Moreover, if $x \neq \widehat{x}_j$ for all $j \in [N]$, then:

1. If $D_f$ is the Hellinger distance, then for any $\varepsilon \in [0, 1]$, we have $\nu_{\text{Hellinger}}^\star(x) = 1 - (1 - \varepsilon)^2$.

2. If $D_f$ is the Pearson's $\chi^2$-divergence, then for any $\varepsilon \in \mathbb{R}_+$, we have $\nu_{\chi^2}^\star(x) = 1 - (1 + \varepsilon)^{-1}$.

3. If $D_f$ is the total variation distance, then for any $\varepsilon \in \mathbb{R}_+$, we have $\nu_{\text{TV}}^\star(x) = \varepsilon/2$.

*Proof of Theorem B.3.* The reformulation as a convex program follows directly from the first part of the proof of Theorem 2.3 using the general function $f$, and it is thus omitted. We now proceed to consider the case when $x \notin \widehat{\mathcal{S}}$, and we derive the optimal value $\nu_f^\star(x)$ for each divergence $f$.

1. **Hellinger distance.** Following the same approach as in the proof of Theorem 2.3, we employ the definition of the Hellinger distance to obtain the equivalent minimization problem

$$\mathrm{OPT}_{\text{Hellinger}}^\star = \min_{y \in \Delta_N} \left\{ \sum_{j \in [N]} y_j : \sum_{j \in [N]} \widehat{\nu}_j - \sum_{j \in [N]} \sqrt{\widehat{\nu}_j}\sqrt{y_j} \leq \varepsilon \right\}.$$

Suppose that $\varepsilon \in (0, 1]$. Using a duality argument, we have

$$
\mathrm{OPT}^{\star}_{\mathrm{Hellinger}} = \min_{y \in \Delta_N} \max_{\gamma \geq 0} \left\{ \sum_{j \in [N]} y_j + \gamma \left( \sum_{j \in [N]} \widehat{\nu}_j - \sum_{j \in [N]} \sqrt{\widehat{\nu}_j}\sqrt{y_j} - \varepsilon \right) \right\}
$$

$$
= \max_{\gamma \geq 0} \left\{ \gamma \left( \sum_{j \in [N]} \widehat{\nu}_j - \varepsilon \right) + \min_{y \in \Delta_N} \left\{ \sum_{j \in [N]} y_j - \gamma \sum_{j \in [N]} \sqrt{\widehat{\nu}_j}\sqrt{y_j} \right\} \right\}
$$

$$
\geq \sup_{2 \geq \gamma > 0} \left\{ \gamma \left( \sum_{j \in [N]} \widehat{\nu}_j - \varepsilon \right) + \min_{y \in \Delta_N} \left\{ \sum_{j \in [N]} y_j - \gamma \sum_{j \in [N]} \sqrt{\widehat{\nu}_j}\sqrt{y_j} \right\} \right\}
$$

$$
= \sup_{2 \geq \gamma > 0} \left\{ \gamma \left( \sum_{j \in [N]} \widehat{\nu}_j - \varepsilon \right) - \frac{\gamma^2}{4} \sum_{j \in [N]} \widehat{\nu}_j \right\},
$$

where we have used the optimal solution $y_j^{\star} = \gamma^2 \widehat{\nu}_j / 4$ to arrive at the last equation. The supremum over $\gamma$ admits the optimal solution $\gamma^{\star} = 2(1 - \varepsilon)$. We can thus show that

$$
OPT^{\star}_{\mathrm{Hellinger}} \geq (1 - \varepsilon)^2 \quad \forall \varepsilon \in (0, 1].
$$

The rest of the proof is analogous to the proof of Theorem 2.3.

2. **Pearson's $\chi^2$-divergence.** By definition of the divergence, we obtain

$$
\mathrm{OPT}^{\star}_{\chi^2} = \min_{y \in \Delta_N} \left\{ \sum_{j \in [N]} y_j : \sum_{j \in [N]} \widehat{\nu}_j^2 y_j^{-1} - \sum_{j \in [N]} \widehat{\nu}_j \leq \varepsilon \right\}.
$$

Suppose that $\varepsilon > 0$. Using a duality argument, we have

$$
\mathrm{OPT}^{\star}_{\chi^2} = \min_{y \in \Delta_N} \max_{\gamma \geq 0} \left\{ \sum_{j \in [N]} y_j + \gamma \left( \sum_{j \in [N]} \widehat{\nu}_j^2 y_j^{-1} - \sum_{j \in [N]} \widehat{\nu}_j - \varepsilon \right) \right\}
$$

$$
= \max_{\gamma \geq 0} \left\{ -\gamma \left( \sum_{j \in [N]} \widehat{\nu}_j + \varepsilon \right) + \min_{y \in \Delta_N} \left\{ \sum_{j \in [N]} y_j + \gamma \sum_{j \in [N]} \widehat{\nu}_j^2 y_j^{-1} \right\} \right\}
$$

$$
\geq \sup_{1 \geq \gamma > 0} \left\{ -\gamma \left( \sum_{j \in [N]} \widehat{\nu}_j + \varepsilon \right) + \min_{y \in \Delta_N} \left\{ \sum_{j \in [N]} y_j + \gamma \sum_{j \in [N]} \widehat{\nu}_j^2 y_j^{-1} \right\} \right\}
$$

$$
= \sup_{1 \geq \gamma > 0} \left\{ -\gamma \left( \sum_{j \in [N]} \widehat{\nu}_j + \varepsilon \right) + 2\sqrt{\gamma} \sum_{j \in [N]} \widehat{\nu}_j \right\},
$$

where we have used the optimal solution $y_j^{\star} = \sqrt{\gamma}\widehat{\nu}_j$ to arrive at the last equation. The supremum over $\gamma$ admits the optimal solution $\gamma^{\star} = (1 + \varepsilon)^{-2}$, which implies that

$$
\mathrm{OPT}^{\star}_{\chi^2} \geq (1 + \varepsilon)^{-1} \quad \forall \varepsilon > 0.
$$

The rest of the proof is analogous to the proof of Theorem 2.3.

3. **Total variation distance.** We have

$$
\mathrm{OPT}^{\star}_{\mathrm{TV}} = \min_{y \in \Delta_N} \left\{ \sum_{j \in [N]} y_j : \sum_{j \in [N]} |\widehat{\nu}_j - y_j| + 1 - \sum_{j \in [N]} y_j \leq \varepsilon \right\}.
$$

(a) Nominal measure $\widehat{\nu}^{(1)}$        (b) Nominal measure $\widehat{\nu}^{(2)}$

Figure 1: Approximations of the likelihood $p(x|\theta)$ under two different nominal measures. The approximation offered by the mean-variance ambiguity set is the same for both $\widehat{\nu}^{(1)}$ and $\widehat{\nu}^{(2)}$. In contrast, the approximation offered by the Wasserstein ambiguity set produces a fatter tail under the nominal measure $\widehat{\nu}^{(2)}$, whose support is more spread out.

For any $\varepsilon \geq 0$, the optimal solution $y^\star$ satisfies $y_j^\star \leq \widehat{\nu}_j$, and thus we have

$$\text{OPT}_{\text{TV}}^\star = \min_{y \in \Delta_N} \left\{ \sum_{j \in [N]} y_j : \sum_{j \in [N]} (\widehat{\nu}_j - y_j) + 1 - \sum_{j \in [N]} y_j \leq \varepsilon \right\}$$

$$= \min_{y \in \Delta_N} \left\{ \sum_{j \in [N]} y_j : 2 - 2 \sum_{j \in [N]} y_j \leq \varepsilon \right\} = 1 - \frac{\varepsilon}{2},$$

which finishes the proof for the total variation distance.

These observations complete the proof.      □

### B.3    Comparison of Moment and Wasserstein Ambiguity Sets

In this section, we empirically demonstrate that the approximation using the Wasserstein ambiguity set can capture the tail behavior of the nominal distribution $\widehat{\nu}$ better than the approximation using the moment ambiguity set. To this end, consider the two univariate discrete nominal measures

$$\widehat{\nu}^{(1)} = \frac{1}{2}\delta_{-1} + \frac{1}{2}\delta_1 \qquad \text{and} \qquad \widehat{\nu}^{(2)} = 0.1\delta_{-2} + 0.4\delta_{-\frac{1}{2}} + 0.4\delta_{\frac{1}{2}} + 0.1\delta_2.$$

Notice that both $\widehat{\nu}^{(1)}$ and $\widehat{\nu}^{(2)}$ share the same mean 0 and the same variance 1, and thus we find that

$$\sup_{\nu \in \mathbb{B}_{\text{MV}}(\widehat{\nu}^{(1)})} \nu(x) = \sup_{\nu \in \mathbb{B}_{\text{MV}}(\widehat{\nu}^{(2)})} \nu(x) \qquad \forall x \in \mathcal{X}.$$

However, if we use the Wasserstein ambiguity set $\mathbb{B}_{\text{W}}(\cdot)$, then in general we have

$$\sup_{\nu \in \mathbb{B}_{\text{W}}(\widehat{\nu}^{(1)},\varepsilon)} \nu(x) \neq \sup_{\nu \in \mathbb{B}_{\text{W}}(\widehat{\nu}^{(2)},\varepsilon)} \nu(x).$$

Figure 1 illustrates the approximations $p(x|\theta)$ offered by the optimal value of the optimistic likelihood problem (3) over these two ambiguity sets. If we choose $\widehat{\nu}^{(2)}$ as the nominal measure, we would expect the true distribution $p(\cdot|\theta)$ to be more spread out than when we choose $\widehat{\nu}^{(1)}$. Nevertheless, this structural information is discarded by the moment ambiguity set, and the optimal value of the optimistic likelihood problem is the same for $\widehat{\nu}^{(1)}$ and $\widehat{\nu}^{(2)}$. In contrast, the Wasserstein ambiguity set produces a fatter tail under the nominal measure $\widehat{\nu}^{(2)}$ than under $\widehat{\nu}^{(1)}$, which better reflects the information contained in the nominal distribution.

Interestingly, if $x = 0$, then we have
$$\sup_{\nu \in \mathbb{B}_{\mathrm{MV}}(\widehat{\nu}^{(1)})} \nu(0) = \sup_{\nu \in \mathbb{B}_{\mathrm{MV}}(\widehat{\nu}^{(2)})} \nu(0) = 1.$$
Indeed, consider the family of discrete measures $\{\nu_k\}_{k \in \mathbb{N}_+}$ defined as
$$\nu_k = \left(1 - \frac{1}{k^2}\right) \delta_0 + \frac{1}{2k^2} \left(\delta_k + \delta_{-k}\right) \qquad \forall k \in \mathbb{N}_+.$$
By construction, $\nu_k$ has mean 0 and variance 1, and thus $\{\nu_k\}_{k \in \mathbb{N}_+}$ belong to $\mathbb{B}_{\mathrm{MV}}(\widehat{\nu}^{(1)})$ and attain the optimal value of 1 asymptotically.

## B.4   Approximation of the Log-Likelihood for Multiple Observations

In many cases, the update of the posterior is carried out after observing a batch of $L$ i.i.d. samples $x_1^L \triangleq \{x_1, \ldots, x_L\}$. In this case, the log-likelihood of the data $x_1^L$ can be written as
$$\log p(x_1^L | \theta) = \log \prod_{\ell \in [L]} p(x_\ell | \theta) = \sum_{\ell \in [L]} \log p(x_\ell | \theta).$$
When $p(\cdot | \theta)$ is intractable, we propose the optimistic log-likelihood approximation
$$\log p(x_1^L | \theta) \approx \sup_{\nu \in \mathbb{B}_\theta(\widehat{\nu}_\theta)} \sum_{\ell \in [L]} \log \nu(x_\ell) \tag{B.4}$$
for some ambiguity set $\mathbb{B}_\theta(\widehat{\nu}_\theta)$ defined below. Note that the optimistic log-likelihood approximation (B.4) follows the spirit of the optimistic likelihood approximation (3).

Because the $\log$ function attains $-\infty$ at 0, we need to restrict ourselves to a subset of $\mathcal{M}(\mathcal{X})$ over which the objective function of (B.4) is well-defined. For any batch data $x_1^L$, we denote by $\mathcal{M}_{x_1^L}(\mathcal{X})$ the set of measures supported on $\mathcal{X}$ with positive mass at any $x_\ell \in x_1^L$, that is,
$$\mathcal{M}_{x_1^L}(\mathcal{X}) = \{\nu \in \mathcal{M}(\mathcal{X}) : \nu(x_\ell) > 0 \ \forall \ell \in [L]\}.$$

We first establish the upper semicontinuity of the objective function in (B.4).

**Lemma B.4** (Upper semicontinuity). *For any batch data $x_1^L$, the functional $G(\nu) = \sum_{\ell \in [L]} \log \nu(x_\ell)$ is upper semicontinuous over $\mathcal{M}_{x_1^L}(\mathcal{X})$.*

*Proof.* Let $\{\nu_k\}_{k \in \mathbb{N}_+}$ be a sequence of probability measures in $\mathcal{M}_{x_1^L}(\mathcal{X})$ converging weakly to $\nu \in \mathcal{M}_{x_1^L}(\mathcal{X})$. We have
$$\limsup_{k \to \infty} G(\nu_k) = \limsup_{k \to \infty} \sum_{\ell \in [L]} \log \nu_k(x_\ell) = \sum_{\ell \in [L]} \log \left(\limsup_{k \to \infty} \nu_k(x_\ell)\right) \leq \sum_{\ell \in [L]} \log \nu(x_\ell) = G(\nu),$$
where the first and last equalities are from the definition of $G$, the second equality is from the continuity of the $\log$ function over $\mathcal{M}_{x_1^L}(\mathcal{X})$, and the inequality is due to the upper semicontinuity of the function $F(\nu) = \nu(x)$ established in Lemma A.1. This completes the proof. $\square$

Given batch data $x_1^L$, we now consider the Wasserstein ambiguity set centered at the nominal distribution $\widehat{\nu}$,
$$\mathbb{B}_{\mathrm{W}}(\widehat{\nu}, \varepsilon) = \{\nu \in \mathcal{M}_{x_1^L}(\mathcal{X}) : \mathrm{W}(\nu, \widehat{\nu}) \leq \varepsilon\},$$
where the dependence on $\theta$ and $x_1^L$ has been made implicit to avoid clutter.

**Theorem B.5** (Optimistic log-likelihood; Wasserstein ambiguity). *Suppose that Assumption ?? holds. For any batch data $x_1^L$ and radius $\varepsilon > 0$, the optimistic log-likelihood problem (B.4) under the Wasserstein ball $\mathbb{B}_{\mathrm{W}}(\widehat{\nu}, \varepsilon)$ is equivalent to the finite convex program*

$$\sup_{\nu \in \mathbb{B}_{\mathrm{W}}(\widehat{\nu}, \varepsilon)} \sum_{\ell \in [L]} \log \nu(x) = \begin{cases} \max & \sum_{\ell \in [L]} \log \left(\sum_{j \in [N]} T_{j\ell}\right) \\ \mathrm{s.t.} & T \in \mathbb{R}_+^{N \times L}, \ \sum_{\substack{j \in [N] \\ \ell \in [L]}} d(\widehat{x}_j, x_\ell) T_{j\ell} \leq \varepsilon \\ & \sum_{\ell \in [L]} T_{j\ell} \leq \widehat{\nu}_j \qquad \forall j \in [N]. \end{cases} \tag{B.5}$$

*Proof.* We first combine the fact that the logarithm is strictly increasing with the proof of Proposition 4.2 to show that there is an optimal measure $\nu_{\mathrm{W}}^{\star}$ that is supported on $\mathrm{supp}(\nu_{\mathrm{W}}^{\star}) \subseteq \mathrm{supp}(\widehat{\nu}) \cup x_1^L$, a finite set of cardinality $N + L$. Notice that the existence of this optimal measure is guaranteed by the upper semicontinuity of the objective function established in Lemma B.4 and the weak compactness of $\mathbb{B}_{\mathrm{W}}(\widehat{\nu}, \varepsilon)$ established in [8, Proposition 3]. The details of this step are omitted for brevity.

Since the optimal measure is supported on $\mathrm{supp}(\widehat{\nu}) \cup x_1^L$, it suffices to consider measures of the form

$$\nu = \sum_{j\in[N]} y_j \delta_{\widehat{x}_j} + \sum_{\ell\in[L]} z_\ell \delta_{x_\ell}$$

for some $y \in \mathbb{R}_+^N$, $z \in \mathbb{R}_+^L$ satisfying $\sum_{j\in[N]} y_j + \sum_{\ell\in[L]} z_\ell = 1$. Using the Definition 4.1 of the type-1 Wasserstein distance, we can rewrite the optimistic log-likelihood problem over the Wasserstein ball $\mathbb{B}_{\mathrm{W}}(\widehat{\nu}, \varepsilon)$ as the convex program

$$
\begin{aligned}
\sup \quad & \sum_{\ell\in[L]} \log(z_\ell) \\
\mathrm{s.\,t.} \quad & y \in \mathbb{R}_+^N,\ z \in \mathbb{R}_+^L,\ \lambda \in \mathbb{R}_+^{N\times(N+L)} \\
& \sum_{j\in[N]}\sum_{j'\in[N]} d(\widehat{x}_j, \widehat{x}_{j'})\lambda_{jj'} + \sum_{j\in[N]}\sum_{\ell\in[L]} d(\widehat{x}_j, x_\ell)\lambda_{j(N+\ell)} \leq \varepsilon \\
& \sum_{j'\in[N+L]} \lambda_{jj'} = \widehat{\nu}_j && \forall j \in [N] \\
& \sum_{j\in[N]} \lambda_{jj'} = y_j && \forall j' \in [N] \\
& \sum_{j\in[N]} \lambda_{jj'} = z_{j'-N} && \forall j' \in [N+L]\backslash[N] \\
& \sum_{j\in[N]} y_j + \sum_{\ell\in[L]} z_\ell = 1.
\end{aligned}
$$

By letting $T_{j\ell} = \lambda_{j(N+\ell)}$ and eliminating the redundant components of $\lambda$, we obtain the desired reformulation. This completes the proof. $\qquad\square$