[Reviews · NeurIPS 2019]

Reviewer 1



Detailed Comments: Originality: To the best of my knowledge, the distributional optimistic approach to approximating the likelihood function, based on constructing an uncertainly set and picking the most likely distribution in this set, is a novel and interesting idea. The idea borrows from the robust optimization community as well as the principle of optimism under uncertainty and is intuitively appealing. Clarity: The paper is for the most part written well and is well organized. Quality/Significance: To the best of my knowledge, the mathematical analysis and proofs are correct. The paper importantly demonstrates that for several important choices of uncertainty sets (e.g., KL and Wasserstein divergences), the optimistic likelihood formulation reduces to a convex optimization problem. Moreover, in the cases of KL and Wasserstein, the paper shows that applying this technique within the ELBO problem for posterior inference has good theoretical asymptotic guarantees. The above two classes of results on reformulations and asymptotic properties are the most important results to establish for this type of methodology, and it appears that the paper does a good job in establishing these results. Finally, while the numerical experiments are largely illustrative on toy datasets, they do also provide some justification for applying this methodology for posterior inference.

Reviewer 2



Overall, I found this paper to be a nice read. It lays out the motivation for the problem and then illustrates how one can apply the idea for various different notions of a "close distribution," e.g., KL-divergence, Wasserstein metric, and distributions that match the first and second empirical moments. One strange thing about this approach is that the optimistic probabilities found at the end may not integrate to 1 (for example, the kernel density estimator will integrate to 1). For this reason, it doesn't appear the optimistic likelihood is a likelihood in any traditional sense. Because of this property, I would like to understand better how this new sense of likelihood behaves. Does it relate to any other already studied notions of convergence? E.g., there are results that kernel density estimates will converge to the true density under certain regularity assumptions as the number of samples grows and the bandwidth decrease. Is some analog true here? One area I found a bit confusing in the paper was how this could be actually used to solve the problem of Bayesian inference referenced at the beginning of the paper. For example, in section 6.1, how does one come up with the hat{v_i}? Does one have to know how to sample from p to produce these distributions? While the ideas presented in the paper are interesting, I'm still unsure whether this approach is better than simpler methods like the kernel density estimation. I would be more convinced if this method were tried on more practical models, e.g., for those where ABC were necessary. Detailed comments: L74: Should say "denoising." L127: Should the v* be v*_{KL}? L133-134: What's e in e^T y = 1? Figure 1: Is it true that p(x) is unbounded near the poles -1 and 1? Is this desired? L224: Is it clear that this approximation really is an approximation? By the looks of Figure 1, it appears the density can be unbounded near x_i. L269-275: How does one construct hat{v}_i in this case? Originality: The paper has a new approach for fitting posterior distributions in a non-parametric fashion. One limitation is that the distributions must be discrete, and they can be expensive to use as the dimension m and data size m increase. Quality: The paper appears to be correct, although I have not checked the proofs from Section 5. Clarity: The paper is quite clear and nice to read. There are a few parts where some details are omitted (see the comments above) but overall it was easy to follow. Significance: This is the biggest area preventing a higher score on this work. Based on the limited empirical examples, fact that Theta must be finite, and fact that the optimistic likelihood isn't a true probability distribution, I am unsure whether this method is truly better than something simpler.

Reviewer 3



* Line 55-63 * is confusing. Why p(\dot|\theta) represents a discrete distribution? For the classification task the authors mentioned, p(x|\theta_i) can be supported on some absolutely continuous measure (w.r.t Lebesgue measure in finite dimension space). One possible explanation is the observation \hat{x} is discrete, so we use the empirical measure of \hat{x} as the base measure, which seems consistent with the following paper (especially Assumption 2.2). But the authors should make the meaning of notation p(\dot|\theta) clear. *Line 67-76* the authors connect their proposed optimistic likelihood estimation with some existing optimum in the face of uncertainty method. However, these methods always consider the optimistic feedback (e.g. UCB in bandits, planning and Bayesian optimization) but not the optimistic likelihood. Can the authors comment more on this? *Theorem 2.4* missing definition of *e*. The authors only discuss the problem when we have some observation S and want to get the likelihood of one point x inside or outside S, but how to deal with the situation we want to simultaneously get the likelihood of several evaluation points (likelihood for each observation, not the sum of log-likelihood just in Appendix B.4)? This can be simply handled with some traditional methods like KDE and ABC methods introduced in the Section 1. And in real worlds, there exists such cases, for example when we approximate the ELBO with monte carlo estimators, we need to evaluate the likelihood from the samples of q at the same time. We cannot use this optimistic likelihood approximation individually for each sample because this may make the measure unnormalized (because we optimistically assign density to S\cup x in each optimization, if I understand correctly). Overall, this paper proposed a novel optimistic non-parametric likelihood estimation. The authors provide some practical estimators with ambiguity sets constructed by f-divergence, moment conditions and Wasserstein distance and all of the claims from the authors have strong theoretical guarantees. The experiments seem not so supportive.

[Author Response · NeurIPS 2019]

We would like to thank all referees for their appreciation of our results and the useful feedback. Below is our reply.

**Reviewer 3:** Thank you for your encouragement to demonstrate the significance of our findings, in response to which we have run further numerical results (Table 1 below). We will include these results in the final version of the manuscript.

**Reviewer 4:** Please find our responses to your comments below.

*(1) Optimistic probabilities may not add up to* 1. We apologize for the confusion. In problem (3), $x$ is a single new observation that is used for our inference (we extend our setting to multiple observations in Appendix B.4). The optimization problem (3) is over probability measures $\nu \in \mathbb{B}_\theta(\widehat{\nu}_\theta)$, and as such its maximizer is by construction a probability measure. Please note that we do *not* solve problem (3) for all values of $x$ – we solve it once for a single observation $x$ (or once for a batch of observations $x$, as discussed in Appendix B.4). We will clarify our intentions in the revised version of the manuscript; thank you!

*(2) One area I found a bit confusing in the paper was how this could be actually used to solve the problem of Bayesian inference referenced at the beginning of the paper.* Thank you. We do indeed assume that we can sample from the conditional distribution $p(\cdot|\theta)$, which is a common assumption in Bayesian statistics and stochastic approximations. We apologize for the lack of clarity in the definition of $\widehat{\nu}_i$. In the numerical section, we construct $\widehat{\nu}_i$ as described in Assumption 5.2 to ensure the convergence guarantees of the ELBO problem. We will elaborate more in the final version.

*(3) Experimental results are not convincing.* We have run further numerical results (see Table 1 below), which we will include in the manuscript. We hope that this gives further support to our methods. Thank you for this suggestion!

You have also provided some more detailed suggestions. We will implement all of these in the revision; thank you!

**Reviewer 6:** Please find our responses to your comments below.

*(1) The distribution $p(\cdot|\theta)$.* We apologize for using informal notation. The unknown true distribution $p(\cdot|\theta)$ can be any measure; it does *not* need to be discrete or continuous. The maximizer $\nu^\star$ in our optimistic likelihood problem (3), in contrast, is discrete for all our ambiguity sets. The hope—which is supported by our convergence analysis as well as our numerical results—is that our discrete approximations are close to $p(\cdot|\theta)$. We will clarify this point in the revision.

*(2) Optimistic likelihood estimation vs. optimism in the face of uncertainty.* Our reasons for relating our paper to the optimism in the face of uncertainty literature are twofold. Firstly, we wanted to highlight that an optimistic treatment of ambiguity, which may be counterintuitive to the (distributionally) robust optimization community, has been successfully applied in a different discipline. Secondly, we did not want to claim that we are the first to exercise optimism in the face of ambiguity and give due credit to the existing literature. We will clarify this in the revision, thank you!

*(3) Theorem 2.4 is missing the definition of* e. e denotes the vector of all ones. We will add this, thank you!

*(4) Likelihood of a set $S$ of observations.* This is a very interesting question. In our understanding, maximizing the likelihood of a set $S$ of observations amounts to a multi-objective optimization problem, where we need to assign a 'priority' (e.g., a weight or a lexicographic ordering) to the probability assigned to each data point $x \in S$. To us, the most natural choice is to maximize the sum of the log likelihoods as done in Appendix B.4, as it maximizes the likelihood of observing all data points $x \in S$ as a batch. Moreover, $\nu^\star$ that solves problem (B.4) in the appendix is also a probability measure, which can be used to evaluate $\nu^\star(x)$ separately for any $x \in S$. We will elaborate on our intentions in Appendix B.4 in the final version of the manuscript.

*(5) Experimental results are not convincing.* We have run further numerical results (see Table 1 below), which we will include in the manuscript. We hope that this gives further support to our methods. Thank you for this suggestion!

**Additional numerical experiments:** Table 1 extends the classification results for real-life datasets in Section 6.1. Our Python source code and experimental data will be published on Github to ensure reproducibility of our results.

Table 1: Average area under the precision-recall curve for various datasets. Bold numbers highlight the best results.

| | Kernel | Moment | Wasserstein | | Kernel | Moment | Wasserstein |
|---|---|---|---|---|---|---|---|
| Banknote | 99.05 | 99.99 | **100.00** | Housing | 80.75 | 81.89 | **83.02** |
| Blood Transfusion | 66.44 | **71.28** | 69.71 | ILPD | 71.75 | **72.95** | 70.12 |
| Breast Cancer | 98.03 | **99.26** | 97.35 | Ionosphere | 91.15 | 97.05 | **98.96** |
| Climate Model | **93.82** | 81.94 | 93.72 | Mammographic | 84.11 | 86.53 | **88.28** |
| Cylinder | 77.37 | 75.00 | **86.59** | Pima | 80.90 | **82.37** | 80.81 |
| German Credit | 67.84 | **75.50** | 75.47 | QSAR | 84.49 | **90.85** | 90.55 |
| Haberman | 72.88 | 70.20 | **73.26** | Sonar | 87.18 | 83.49 | **94.45** |
| Heart | 79.46 | **86.87** | 77.07 | Thoracic | 58.73 | **64.73** | 59.89 |

[Meta-Review · NeurIPS 2019]

After discussion, the reviewers agreed that the paper was written clearly and the theoretical contributions were sound and interesting; enough so that the paper was worth accepting on those merits. But they also agreed that, despite the rebuttal, there is still uncertainty in the practicality of the algorithm and empirical results. The results (in the paper + rebuttal) appear to be limited to cases where ABC isn't necessary, and the comparison to significantly simpler kernel methods is not really comprehensive. For the final draft, please be sure to add the results from the rebuttal, as well as more thorough evidence that the method performs well against common kernel methods and in settings where ABC is required.